# A LEARNING THEORETIC PERSPECTIVE ON LOCAL EXPLAINABILITY

**Jeffrey Li**[*]
University of Washington
jwl2162@cs.washington.edu

**Vaishnavh Nagarajan**[*]**, Gregory Plumb**
Carnegie Mellon University
vaishnavh@cs.cmu.edu

**Ameet Talwalkar**
Carnegie Mellon University & Determined AI

## ABSTRACT

In this paper, we explore connections between interpretable machine learning and learning theory through the lens of *local approximation* explanations. First, we tackle the traditional problem of *performance generalization* and bound the test-time predictive accuracy of a model using a notion of how locally explainable it is. Second, we explore the novel problem of *explanation generalization* which is an important concern for a growing class of *finite sample-based* local approximation explanations. Finally, we validate our theoretical results empirically and show that they reflect what can be seen in practice.

## 1 INTRODUCTION

There has been a growing interest in interpretable machine learning, which seeks to help people better understand their models. While interpretable machine learning encompasses a wide range of problems, it is a fairly uncontroversial hypothesis that there exists a trade-off between a *model's complexity* and *general notions of interpretability*. This hypothesis suggests a seemingly natural connection to the field of learning theory, which has thoroughly explored relationships between a *function class's complexity* and *generalization*. However, formal connections between interpretability and learning theory remain relatively unstudied.

Though there are several notions of conveying interpretability, one common and flexible approach is to use local approximations. Formally, *local approximation explanations* (which we will refer to as "local explanations") provide insight into a model's behavior as follows: for any black-box model $f \in \mathcal{F}$ and input $x$, the explanation system produces a simple function, $g_x(x') \in \mathcal{G}_{\text{local}}$, which approximates $f$ in a chosen neighborhood, $x' \sim N_x$. Crucially, the freedom to specify both $\mathcal{G}_{\text{local}}$ and $N_x$ grants local explanations great versatility. In this paper, we provide two connections between learning theory and how well $f$ can be approximated locally (i.e. the *fidelity* of local explanations).

Our first result studies the standard problem of *performance generalization* by relating test-time predictive accuracy to a notion of local explainability. As it turns out, our focus on local explanations leads us to unique tools and insights from a learning theory point of view. Our second result identifies and addresses an unstudied – yet important – question regarding *explanation generalization*. This question pertains to a growing class of explanation systems, such as MAPLE (Plumb et al., 2018) and RL-LIM (Yoon et al., 2019), which we call *finite sample-based* local explanations[1]. These methods learn their local approximations using a common finite sample drawn from data distribution $D$ (in contrast to canonical local approximation methods such as LIME (Ribeiro et al., 2016)) and, as a result, run the risk of overfitting to this finite sample. In light of this, we answer the following question: for these explanation-learning systems, how well do the quality of local explanations generalize to data not seen during training?

---

[*]Denotes equal contribution
[1]This terminology is not to be confused with "example-based explanations" where the explanation itself is in the form of data instances rather than a function.

We address these questions with two bounds, which we outline now. Regarding performance generalization, we derive our first main result, Theorem 1, which bounds the expected test mean squared error (MSE) of any $f$ in terms of its MSE over the $m$ samples in the training set, $S = \{(x_i, y_i)\}_{i=1}^m$:

$$\underbrace{\mathbb{E}_{(x,y)\sim D}[(f(x) - y)^2]}_{\text{Test MSE}} \leq \tilde{\mathcal{O}}\Big( \underbrace{\frac{1}{m} \sum_{i=1}^m (f(x_i) - y_i)^2}_{\text{Train MSE}} + \underbrace{\mathbb{E}_{\substack{x\sim D, \\ x'\sim N_x}} \left[(g_{x'}(x) - f(x))^2\right]}_{\text{Interpretability Term (MNF)}} + \underbrace{\rho_S \hat{\mathcal{R}}_S(\mathcal{G}_{\text{local}})}_{\text{Complexity Term}} \Big)$$

Regarding explanation generalization for finite sample-based explanation-learning systems, we apply a similar proof technique to obtain Theorem 2, which bounds the quality of the system's explanations on unseen data in terms of their quality on the data on which the system was trained:

$$\underbrace{\mathbb{E}_{\substack{x\sim D, \\ x'\sim N_x}} \left[(g_{x'}(x) - f(x))^2\right]}_{\text{Test MNF}} \leq \underbrace{\frac{1}{m} \sum_{i=1}^m \mathbb{E}_{x'\sim N_{x_i}} \left[(f(x_i) - g_{x'}(x_i))^2\right]}_{\text{Train MNF}} + \underbrace{\tilde{\mathcal{O}}\left(\rho_S \hat{\mathcal{R}}_S(\mathcal{G}_{\text{local}})\right)}_{\text{Complexity Term}}$$

Before summarizing our contributions, we discuss the key terms and their relationships.

- *Interpretability terms:* The terms involving MNF correspond to Mirrored Neighborhood Fidelity, a metric we use to measure local explanation quality. As we discuss in Section 3, this is a reasonable modification of the commonly used Neighborhood Fidelity (NF) metric (Ribeiro et al., 2016; Plumb et al., 2018). Intuitively, we generally expect MNF to be larger when the neighborhood sizes are larger since each $g_{x'}$ is required to extrapolate farther.

- *Complexity term:* This term measures the complexity of the local explanation system $g$ in terms of (a) the complexity of the local explanation class $\mathcal{G}_{\text{local}}$ and (b) $\rho_S$, a quantity that we define and refer to as the *neighborhood disjointedness factor*. As we discuss in Section 4, $\rho_S$ is a value in $[1, \sqrt{m}]$ (where $m = |S|$) that is proportional to the level of disjointedness of the neighborhoods for points in the sample $S$. Intuitively, we expect $\rho_S$ to be larger when the neighborhoods sizes are smaller since smaller neighborhoods will overlap less.

Notably, both our bounds capture the following key trade-off: as neighborhood widths increase, MNF increases but $\rho_S$ decreases. As such, our bounds are non-trivial only if the neighborhoods $N_x$ can be chosen such that MNF remains small but $\rho_S$ grows slower than $\tilde{\mathcal{O}}(\sqrt{m})$ (since $\hat{\mathcal{R}}_S(\mathcal{G}_{\text{local}})$ typically decays as $\tilde{\mathcal{O}}(1/\sqrt{m})$).

We summarize our main contributions as follows:

(1) We make a novel connection between performance generalization and local explainability, arriving at Theorem 1. Given the relationship between MNF and $\rho_S$, this bound roughly captures that an easier-to-interpret $f$ enjoys better generalization guarantees, a potentially valuable result when reasoning about $\mathcal{F}$ is difficult (e.g. for neural networks). Further, our proof technique may be of independent theoretical interest as it provides a new way to bound the Rademacher complexity of a particular class of *randomized* functions (see Section 4).

(2) We motivate and explore an important generalization question about expected explanation quality. Specifically, we arrive at Theorem 2, a bound for test MNF in terms of train MNF. This bound suggests that practitioners can better guarantee good local explanation quality (measured by MNF) using methods which encourage the neighborhood widths to be wider (see Section 5).

(3) We verify empirically on UCI Regression datasets that our results non-trivially reflect the two types of generalization in practice. First, we demonstrate that $\rho_S$ can indeed exhibit slower than $\tilde{\mathcal{O}}(\sqrt{m})$ growth without significantly increasing the MNF terms. Also, for Theorem 2, we show that the generalization gap indeed improves with larger neighborhoods (see Section 6).

(4) Primarily to aid in our theoretical results, we propose MNF as a novel yet reasonable measure of local explainability. Additionally, we argue that this metric presents a promising avenue for future study, as it may naturally complement NF and offer a unique advantage when evaluating local explanations on "realistic" on-distribution data (see Section 3).

## 2 Related Work

**Interpretability meets learning theory.** Semenova et al. (2020) study the performance generalization of models learned from complex classes when they can be globally well-approximated by simpler (e.g. interpretable) classes. In such cases, their theory argues that if the complex class has many models that perform about as optimally on training data, generalization from the complex class can be more closely bounded using the simpler class's complexity. In our corresponding results, we similarly aim to avoid involving the larger class's complexity. However, we directly study generalization via a model's local explainability, rather than instantiate "complex" and "simple" classes for global approximations. The two are fundamentally different technical problems; standard learning theory results cannot be directly applied as they are done for single-function global approximations.

**Statistical localized regression.** (Fan, 1993; Fan & Gijbels, 1996) are canonical results which bound the squared error of a nonparametric function defined using locally fit models. These local models are both simple (e.g. linear) and similarly trained by weighting real examples with a kernel (i.e. an implied neighborhood). However, each local model is only used to make a prediction at its source point and the theory requires shrinking the kernel width towards 0 as the sample size grows. We instead fit local models *as explanations* for a trained model (i.e. which is considered the "true regression function") and more importantly, care about the fidelity of each local model over whole (non-zero) neighborhoods. Unlike localized regression, this allows us to use uniform convergence to bound test error with empirical and generalization terms. While the previous results do not have empirical terms, the learning rates are exponential in the number of samples.

**Learning Theory.** Another line of related work also studies how to explain generalization of overparameterized classes. As standard uniform convergence on these classes often leads to vacuous bounds, a general approach that has followed from (Nagarajan & Kolter, 2019; Zhang et al., 2017; Neyshabur et al., 2014) has been to study implications of different biases placed on learned models. We study what would happen if an overparameterized model had an unexplored type of bias, one that is inspired by local explainability. Additionally, our work's technical approach parallels another line of existing results which likewise try to apply uniform convergence on separate surrogate classes. This includes PAC-Bayesian bounds, a large family of techniques that come from looking at a stochastic version of a model in parameter space (McAllester, 1998; 2003; Langford & Caruana, 2002; Langford & Shawe-Taylor, 2003). In a different vein, some results in deep learning look at compressed, sparsified, or explicitly regularized surrogates of neural networks (Arora et al., 2018; Dziugaite & Roy, 2017). In our case, the surrogate class is a collection of local explanations.

## 3 Mirrored Neighborhood Fidelity

In order to connect local explanations to generalization, recall that we study a measure of local interpretability which we call "mirrored neighborhood fidelity" (MNF). As we explain below, this quantity comes from a slight modification to an existing metric for local approximation explanations, namely, that of neighborhood fidelity (NF).

To define these quantities, we use the following notation. Let $\mathcal{X}$ be an input space and $D$ be a distribution over $\mathcal{X} \times \mathcal{Y}$ where $\mathcal{Y} \subseteq \mathbb{R}$. Then, let $\mathcal{F}$ be a class of functions $f : \mathcal{X} \to \mathcal{Y}$. For our theoretical results, we specifically assume that $\mathcal{Y}$ is bounded as $\mathcal{Y} = [-B, B]$ for some $B > 0$ (though this does not matter for the remainder of this section). In order to provide local explanations, we need to fix a nearby region around each $x \in \mathcal{X}$. To this end, for any $x$, let $N_x$ correspond to some distribution denoting a *local neighborhood* at $x$ (e.g. typically chosen to be a distribution centered at $x$). For any distribution $N$, we use $p_N(x)$ to denote its density at $x$. Finally, let $\mathcal{G}$ be a class of local explainers $g : \mathcal{X} \times \mathcal{X} \to \mathcal{Y}$ such that for each $x \in \mathcal{X}$, the local explanation $g(x, \cdot) : \mathcal{X} \to \mathcal{Y}$ belongs to a class of (simple) functions (e.g. linear), $\mathcal{G}_{\text{local}}$. For convenience, we denote $g(x, \cdot)$ as $g_x(\cdot)$ and use it to locally approximate $f$ in the neighborhood defined by $N_x$.

The accuracy of the local explanation system $g$ is usually quantified by a term called "neighbhorhood fidelity" which is defined as follows (Ribeiro et al., 2016; 2018; Plumb et al., 2018; 2020):

$$\mathsf{NF}(f, g) := \mathbb{E}_{x \sim D}\left[\mathbb{E}_{x' \sim N_x}\left[(f(x') - g_x(x'))^2\right]\right].$$

To verbally interpet this, let us call $x$ as the "source" point which gives rise to a local explanation $g_x(\cdot)$ and $x'$ the "target" points that we try to fit using $g$. To compute $\mathsf{NF}(f, g)$, we need to do the

following: for each source point $x$, we first compute the average error in the fit of $g_x(\cdot)$ over draws of nearby target points $x' \sim N_x$; then, to get a more overall measure of $g$'s quality, we globally average this error across draws of the source point $x \sim D$.

Now, to define MNF, we take the same expression as for NF but swap $x$ and $x'$ within the innermost expectation (without modifying how each variable is distributed). In other words, we now sample *a target point* $x$ from $D$ and sample *source points* $x'$ from a distribution over points near a given $x$. Since this local distribution is over source points rather than target points, just for the sake of distinguishing, we'll refer to this as a *mirrored* neighborhood distribution and denote it as $N_x^{\text{mir}}$. We define MNF more formally below, following which we explain how to better understand it:

**Definition 3.1.** *(Mirrored Neighborhood Fidelity)* *We define* MNF $: \mathcal{F} \times \mathcal{G} \to \mathbb{R}$ *as*

$$\mathsf{MNF}(f, g) := \mathbb{E}_{x \sim D}\left[\mathbb{E}_{x' \sim N_x^{\text{mir}}}\left[(f(x) - g_{x'}(x))^2\right]\right].$$

*and with an abuse of notation, we let* $\mathsf{MNF}(f, g, x) := \mathbb{E}_{x' \sim N_x^{\text{mir}}}\left[(f(x) - g_{x'}(x))^2\right]$.

**Understanding MNF.** It is helpful to parse the expression for MNF in two different ways. First, we can think of it as measuring the error in approximating every target point $x \in \mathcal{X}$ through a *randomized* locally-approximating function $g_{x'}(\cdot)$, where $x'$ is randomly drawn from the neighborhood $N_x^{\text{mir}}$. A second way to parse this is in a manner similar to how we parsed NF. To do this, first we note that the expectations in MNF can be swapped around and rewritten as:

$$\mathsf{MNF}(f, g) = \mathbb{E}_{x' \sim D^\dagger}\left[\mathbb{E}_{x \sim N_{x'}^\dagger}\left[(f(x) - g_{x'}(x))^2\right]\right],$$

where $D^\dagger$ and $N_{x'}^\dagger$ are suitably defined distributions (derived in Appendix A) that can be thought of as modified counterparts of $D$ and $N_{x'}^{\text{mir}}$ respectively. With this rewritten expression, one can read MNF like NF: for each source point (now $x'$), we compute the average error in the fit of the corresponding local function ($g_{x'}(\cdot)$) over target points ($x$) drawn from the source's local neighborhood ($N_{x'}^\dagger$); this error is then globally averaged over different draws of the source point ($x' \sim D^\dagger$).

**Why MNF?** While both NF and MNF are closely related measures of local explanation quality on $f$, studying MNF allows us to make connections between local explainability and different notions of generalization (Sections 4 and 5). At a high-level, our results don't apply to NF because the overall distribution of the points that are "fit" to calculate NF (i.e., the target points) is not the same as the test data distribution $D$, instead being $D$ perturbed by $N_x$. Rather, we prefer this target distribution to be $D$, which is the case for MNF, for us to be able to neatly bound test-time quantities via a local explainability term. Otherwise, we would have to end up introducing many cumbersome terms.

Furthermore, MNF may also be of practical interest to the interpretability community, as it potentially offers a unique advantage over NF when the intended usage of local explanations is centered around understanding how $f$ works on the specific learning task it was trained on. Specifically, we present as an exploratory argument that selecting the target point distribution to be $D$ rather than $D$ perturbed by $N_x$ (as for NF) may better emphasize the ability of $g$ to approximate $f$ at *realistic* input points. This is relevant for ML (and deep learning especially) because (a) high-dimensional datasets often exhibit significant feature dependencies and adherence to lower dimensional manifolds; (b) $f$ can often be highly unpredictable and unstable when extrapolating beyond training data. Thus, when one measures NF with standard neighborhood choices that ignore feature dependencies (e.g. most commonly $N_x = \mathcal{N}(x, \sigma I)$), the resulting target distribution may concentrate significantly on regions that are non-relevant to the original learning task at hand. As we illustrate in a toy example, this can lead to overemphasis on fitting noisy off-manifold behavior, deteriorating the fit of explanations relative to task-relevant input regions (we defer a more detailed discussion of this point and of other trade-offs between NF and MNF to Appendix A).

## 4 GENERALIZATION OF MODEL PERFORMANCE VIA MNF

The generalization error of the function $f$ is typically bounded by some notion of the representational complexity of $\mathcal{F}$. While standard results bound complexity in terms of parameter count, there is theoretical value in deriving bounds involving other novel terms. By doing so, one might understand how regularizing for those terms can affect the representation capacity, and in turn, the generalization

error of $f$. Especially when $f$'s complexity may be intractable to bound on its own, introducing these terms provides a potentially useful new way to understand $f$'s generalization.

Here specifically, we are interested in establishing a general connection between the representation complexity and the local explainability of *any* $f$. This naturally requires coming up with a notion that appropriately quantifies the complexity of $\mathcal{G}$, which we discuss in the first part of this section. As we shall see, $\mathcal{G}$'s complexity can be expressed in terms of $\mathcal{G}_{\text{local}}$, which is generally less complex and more amenable to standard analysis than $\mathcal{F}$ in practical settings where interpretability is desired. In the second part, we then relate this quantity to the generalization of $f$ to derive our first main result.

**Key technical challenge: bounding the complexity of $\mathcal{G}$.** The overall idea behind how one could tie the notions of generalization and local explainability is fairly intuitive. For example, consider a simplified setting where we approximate $f$ by dividing $\mathcal{X}$ into $K$ disjoint pieces, i.e. neighborhoods, and then approximating each neighborhood via a simple (say, linear) model. Then, one could bound the generalization error of $f$ as the sum of two quantities: first, the error in approximating $f$ via the piecewise linear model, and second, a term involving the complexity of said piecewise model. It is straightforward to show that this complexity term grows polynomially with the piece-count, $K$, and also the complexity of the simple local approximator (see Appendix C.0.1). Similarly, one could hope to bound the generalization error of $f$ in terms of $\mathsf{MNF}(f, g)$ and the complexity of $\mathcal{G}$. However, the key challenge here is that the class $\mathcal{G}$ is a much more complex class than the above class of piecewise linear models. For example, a straightforward piece-count-based complexity bound would be infinitely large since there are effectively infinitely many unique pieces in $g$.

Our core technical contribution here is to bound the (Rademacher) complexity of $\mathcal{G}$ in this more flexible local explanation setting. At a high level, the resulting bound (to be stated shortly) grows inversely with "the level of overlap" between the neighborhoods $\{N_x^{\text{mir}} | x \in \mathcal{X}\}$, quantified as:

**Definition 4.1.** *(Neighborhood disjointedness factor) Given a dataset $S \in (\mathcal{X} \times \mathcal{Y})^m$, we define the neighborhood disjointedness factor $\rho_S$ as*

$$\rho_S := \int_{x' \in \mathcal{X}} \sqrt{\frac{1}{m} \sum_{i=1}^{m} (p_{N_{x_i}^{\text{mir}}}(x'))^2} dx'$$

**Understanding the disjointedness factor.** $\rho_S$ can be interpreted as bounding the "effective number" of pieces induced by the set of neighborhood distributions $\{N_x^{\text{mir}} | x \in \mathcal{X}\}$. This turns out to be a quantity that lies in $[1, \sqrt{m}]$ (shown formally in Appendix Fact B.1). To reason more intuitively about this quantity, it is helpful to consider its behavior in extreme scenarios. First, consider the case where $N_x^{\text{mir}}$ is the same distribution (say $N$) regardless of $x$; i.e., neighborhoods are completely overlapping. Then, $\rho_S = \int_{x' \in \mathcal{X}} (p_N(x')) dx' = 1$. In the other extreme, consider if neighborhoods centered on the training data are all disjoint with supports $\mathcal{X}_1, \ldots, \mathcal{X}_{|S|}$. Here, the integral splits into $m$ summands as: $\rho_S = \sum_{i=1}^{m} \int_{x' \in \mathcal{X}_i} \frac{1}{\sqrt{m}} p_{N_{x_i}^{\text{mir}}}(x') dx' = \sqrt{m}$. Thus, $\rho_S$ grows from $1$ to $\sqrt{m}$ as the level of overlap between the neighborhoods $N_{x_1}^{\text{mir}}, \ldots, N_{x_{|S|}}^{\text{mir}}$ reduces. For intuition at non-extreme values, we show in Appendix B.2 that in a simple setting, $\rho = \sqrt{m^{1-k}}$ (where $0 \leq k \leq 1$) if every neighborhood is just large enough to encompass a $1/m^{1-k}$ fraction of the mass of $D$.

**Rademacher complexity of $\mathcal{G}$.** We now use $\rho_S$ to bound the empirical Rademacher complexity of $\mathcal{G}$. Recall that the empirical Rademacher complexity of a function class $\mathcal{H}$ consisting of $h : \mathcal{X} \to \mathbb{R}$ is defined as $\hat{\mathcal{R}}_S(\mathcal{H}) := \mathbb{E}_{\vec{\sigma}} \left[ \frac{1}{m} \sup_{h \in \mathcal{H}} \sigma_i h(x_i) \right]$, where the $\sigma_i$'s are i.i.d. and drawn uniformly from $\{-1, 1\}$. Roughly, this captures the complexity of $\mathcal{H}$ by measuring how well it can fit random labels. Standard results allow us to then use $\hat{\mathcal{R}}_S(\mathcal{H})$ to bound the generalization error for $h \in \mathcal{H}$.

Now, in order to define the Rademacher complexity of $\mathcal{G}$ (which consists of a different kind of functions whose domain is $\mathcal{X} \times \mathcal{X}$ instead of $\mathcal{X}$), it is useful to think of $g$ as *a randomized function*. Specifically, at any target point $x$, the output of $g$ is a random variable $g_{x'}(x)$ where the randomness comes from $x' \sim N_x^{\text{mir}}$. Then, in Lemma 4.1, we take this stochasticity into account to define and bound the complexity of $\mathcal{G}$. To keep our statement general, we consider a generic loss function $L : \mathbb{R} \times \mathbb{R} \to \mathbb{R}$ (e.g., the squared error loss is $L(y, y') = (y - y')^2$). Indeed, whenever $L$ satisfies a standard Lipschitz assumption, we can bound the complexity of $\mathcal{G}$ (composed with the loss function $L$) in terms of $\rho_S$, the complexity of $\mathcal{G}_{\text{local}}$, and the Lipschitzness of $L$:

**Lemma 4.1.** *(see Appendix Lemma D.1 for full, precise statement) Let $L(\cdot, y')$ be a c-Lipschitz function w.r.t. $y'$ in that for all $y_1, y_2 \in [-B, B]$, $|L(y_1, y') - L(y_2, y')| \le c|y_1 - y_2|$. Then, the empirical Rademacher complexity of $\mathcal{G}$ under the loss function $L$ is defined and bounded as:*

$$\hat{\mathcal{R}}_S(L \circ \mathcal{G}) := \mathbb{E}_{\vec{\sigma}}\left[\sup_{g \in \mathcal{G}} \frac{1}{m} \sum_i^m \sigma_i \mathbb{E}_{x' \sim N_{x_i}^{\mathrm{mir}}}[L(g_{x'}(x_i), y_i)]\right] \le \mathcal{O}\left(c\rho_S \hat{\mathcal{R}}_S(\mathcal{G}_{\mathrm{local}}) \cdot \ln m\right)$$

*where $\vec{\sigma}$ is uniformly distributed over $\{-1, 1\}^m$.*

We note that the proof technique employed here may be of independent theoretical interest as it provides a novel way to bound the complexity of a (particular type of) randomized function. Although techniques like PAC-Bayes provide ways to bound the complexity of randomized functions, they only apply to functions where the randomization comes from stochasticity in the parameters, which is not the case here.

**Main result.** With the above key lemma in hand, we are now ready to prove our main result, which bounds the generalization error of $f$ in terms of the complexity of $\mathcal{G}$, thereby establishing a connection between model generalization and local explainability.

**Theorem 1.** *(see Appendix Theorem 3 for full, precise statement) With probability over $1 - \delta$ over the draws of $S = \{(x_1, y_1), \ldots, (x_m, y_m)\} \sim D^m$, for all $f \in \mathcal{F}$ and for all $g \in \mathcal{G}$, we have (ignoring $\ln 1/\delta$ factors):*

$$\mathbb{E}_{(x,y)\sim D}[(f(x) - y)^2] \le \frac{4}{m} \sum_{i=1}^m (f(x_i) - y_i)^2 + 2\underbrace{\mathbb{E}_{x \sim D}[\mathbb{E}_{x' \sim N_x^{\mathrm{mir}}}[(f(x) - g_{x'}(x))^2]]}_{\mathsf{MNF}(f,g)}$$

$$+ \frac{4}{m} \sum_{i=1}^m \underbrace{\mathbb{E}_{x' \sim N_x^{\mathrm{mir}}}[(f(x_i) - g_{x'}(x_i))^2]}_{\mathsf{MNF}(f,g,x_i)} + \mathcal{O}(B\rho_S \hat{\mathcal{R}}_S(\mathcal{G}_{\mathrm{local}}) \ln m).$$

This result decomposes the test error of $f$ into four quantities. The first quantity corresponds to the training error of $f$ on the training set $S$. The second and the third correspond to the MNF of $f$ with respect to $g$, computed on test and training data respectively. The fourth and final quantity corresponds to a term that bounds the complexity of $\mathcal{G}$ in terms of the "disjointedness factor" $\rho_S$ and the complexity of the local function class $\mathcal{G}_{\mathrm{local}}$.

**Takeaway.** A key aspect of this bound is the trade-off that it captures when varying neighborhood widths. Consider shrinking the neighborhood widths to smaller and smaller values, in turn creating less and less overlap between the neighborhoods of the training examples in $S$. Then, on the one hand, we'd observe that the complexity term (the fourth term on the R.H.S) increases. Specifically, since $\hat{\mathcal{R}}_S(\mathcal{G}_{\mathrm{local}})$ typically scales as $\mathcal{O}(1/\sqrt{m})$, as we go from the one extreme of full overlap to the other extreme of complete disjointedness, the complexity term increases from $\mathcal{O}(1/\sqrt{m})$ to $\mathcal{O}(1)$. At this upper extreme, the bound becomes trivial, as such a constant upper bound would directly follow from just the $\mathcal{O}(1)$ bounds assumed on $\mathcal{Y}$. On the other hand, as the widths decrease, the fidelity terms (the second and third terms) would likely *decrease* – this is because the simple functions in $\mathcal{G}_{\mathrm{local}}$ would find it increasingly easier to approximate $f$ on the shrinking neighborhoods.

This trade-off is intuitive. A function $f$ that is hardly amenable to being fit by local explanations would require extremely tiny neighborhoods for $\mathcal{G}_{\mathrm{local}}$ to locally approximate it (i.e. make the MNF terms small). For example, in an extreme case, when the neighborhoods $N_x^{\mathrm{mir}}$ are set be point masses at $x$, it is trivially easy to find $g_{x'}(\cdot) \in \mathcal{G}_{\mathrm{local}}$ with no approximation error. Thus, the complexity term would be too large in this case, implying that a hard-to-interpret $f$ results in bad generalization. On the other hand, when $f$ is easier to interpret, then we'd expect it to be well-approximated by $\mathcal{G}_{\mathrm{local}}$ even with wider neighborhoods. This allows one to afford smaller values for *both* the complexity and MNF terms. In other words, an easy-to-interpret $f$ enjoys better generalization guarantees.

**Caveats.** Our bound has two limitations worth noting. First, for high-dimensional datasets (like image datasets), practical choices of $N_x^{\mathrm{mir}}$ can lead to almost no overlap between neighborhoods, rendering the bound trivial in practice. This potentially poor dimension-dependence is a caveat similarly shared by bounds for non-parametric local regression, whereby increasing $d$ results in an exponential increase in the required sample size (Fan, 1993; Fan & Gijbels, 1996). Nevertheless, we

show later in our experiments that for lower-dimensional datasets and for practical choices of $N_x^{\text{mir}}$, there *can be* sufficient neighborhood overlap to achieve values of $\rho_S$ that are $o(\sqrt{m})$.

A second caveat is that the second quantity, $\mathsf{MNF}(f, g)$, requires *unlabeled* test data to be computed, which may be limiting if one is interested in numerically computing this bound in practice. It is however possible to get a bound without this dependence, although only on the test error of $g$ rather than $f$ (see Appendix Theorem 4). Nevertheless, we believe that the above bound has theoretical value in how it establishes a connection between the interpretability of $f$ and its generalization.

## 5 GENERALIZATION OF LOCAL EXPLAINABILITY

We now turn our attention to a more subtle kind of generalization that is both unstudied yet important. Typically, the way $g_{x'}$ is learned at any source point $x'$ is by fitting a finite set of points sampled near $x'$, with the hope that this fit generalizes to unseen, neighboring target points. Naturally, we would want to ask: how well do the explanations $g_{x'}$ themselves generalize in this sense?

This question is straightforward to answer in settings like for LIME (Ribeiro et al., 2016). For instance, assume that we learn $g_{x'}$ by sampling a set $S_{x'}$ of points from a Gaussian centered at $x'$, and that we care about the fit of $g_{x'}$ generalizing to the same Gaussian. Here, we can make a standard arugment based on Rademacher complexity to bound the error of $g_{x'}$ on the Gaussian by its training error on $S_{x'}$ and $\hat{\mathcal{R}}_{S_{x'}}(\mathcal{G}_{\text{local}})$. We can apply the same argument individually for the explanation at each source point $x'$, because we have a fresh dataset $S_{x'}$ to generate each $g_{x'}$. If one has the resources to sample more points to create larger $S_{x'}$, then these bounds will also naturally tighten.

However, consider *finite sample-based* local explanation settings like MAPLE (Plumb et al., 2018) and RL-LIM (Yoon et al., 2019) where the training procedure is vastly different: in these procedures, the goal is to learn local explanations $g_{x'}$ in a way that is sensitive to the local structure of the (unknown) underlying data distribution $D$. So, instead of fitting the $g_{x'}$ to samples drawn from an arbitrarily defined (e.g. Gaussian) distribution, here one first draws a finite sample $S$ from $D$ and then labels it using $f$. Then, across all $x' \in \mathcal{X}$, one reuses the same dataset $S$, but learns each $g_{x'}$ on a correspondingly *reweighted* version of $S$ (typically, points in $S$ that are nearer to $x'$ are weighted more heavily). Contrast this with the former setting, where for each $x'$, one has access to a *fresh* dataset (namely, $S_{x'}$) to learn $g_{x'}$. This distinction then makes it interesting to wonder when the reuse of a common dataset $S$ could cause the explanations to generalize poorly.

Motivated by this question, we present Theorem 2. By using Lemma 4.1, we provide a bound on the "test MNF" (which corresponds to the fit of $g_{x'}$ on the unseen data averaged across $D$) in terms of the "train MNF" (which corresponds to the fit of $g_{x'}$ on $S$) and the complexity term from Lemma 4.1. We must however caution the reader that this theorem does *not* answer the exact question posed in the above paragraph; it only addresses it indirectly as we discuss shortly.

**Theorem 2.** *(see Appendix Theorem 2-full for full, precise statement) For a fixed function $f$, with high probability $1 - \delta$ over the draws of $S \sim D^m$, for all $g \in \mathcal{G}$, we have (ignoring $\ln 1/\delta$ factors):*

$$\underbrace{\mathbb{E}_{\substack{x \sim D, \\ x' \sim N_x}} \left[ (f(x) - g_{x'}(x))^2 \right]}_{\text{Test MNF } i.e.,\ \mathsf{MNF}(f,g)} \leq \underbrace{\frac{1}{m} \sum_{i=1}^{m} \mathbb{E}_{x' \sim N_{x_i}^{\text{mir}}} \left[ (f(x_i) - g_{x'}(x_i))^2 \right]}_{\text{Train MNF}} + \mathcal{O}(\rho_S \hat{\mathcal{R}}_S(\mathcal{G}_{\text{local}}) \ln m).$$

**Understanding the overall bound.** We first elaborate on how this bound provides an (indirect) answer to our question about how well explanations generalize. Consider a procedure like MAPLE that learns $g$ using the finite dataset $S$. For each $x' \in \mathcal{X}$, we would expect this procedure to have learned a $g_{x'}$ that fits well on at least those target points $x$ in $S$ that are near $x'$. In doing so, it's reasonable to expect the training procedure to have implicitly controlled the "train MNF" term. The reasoning for this is that the train MNF computes the error in the fit of $g_{x'}$ on $S$ for different values of $x'$, and sums these up in a way that errors corresponding to nearby values of $(x, x')$ are weighted more (i.e., the weight is given by $p_{N_x^{\text{mir}}}(x')$). Now, our bound suggests that when the train MNF is minimized, this carries over to test MNF too, *provided the complexity term is not large*. That is, we can say that the fit of $g_{x'}$ generalizes well to unseen, nearby target points $x$ that lie outside of $S$.

**The indirectness of our result.** Existing finite sample-based explainers do not explicitly minimize the train MNF term (e.g., MAPLE minimizes an error based upon NF). However, as argued above, they have implicit control over train MNF. Hence, our bound essentially treats MNF as a surrogate for reasoning about the generalization of the explanations learned by an arbitrary procedure. As such, our bound does *not* comment on how well the exact kind of fidelity metric used during training generalizes to test data. Nevertheless, we hope that this result offers a concrete first step towards quantifying the generalization of local explanations. Furthermore, we also note that one could also imagine a novel explanation-learning procedure that does explicitly minimize the train MNF term to learn $g$; in such a case our bound would provide a *direct* answer to how well its explanations generalize. Indeed, we derive such a theoretically-principled algorithm in Appendix A.

**Takeaway.** While the above bound captures a similar trade-off involving neighborhood width as Theorem 1, it is worth pausing to appreciate the distinct manner in which this trade-off arises here. In particular, when the width is too small, we know that the complexity term approaches $\mathcal{O}(1)$ and generalization is poor. Intuitively, this is because in this case, the procedure for learning $g_{x'}$ would have been trained to fit very few datapoints from $S$ that happened to have fallen in the small neigbhorhood of $x'$. On the other hand, when the neighborhoods are large, many datapoints from $S$ are likely to fall in each neighborhood, thus allowing each $g_{x'}$ to be trained on an effectively larger dataset. However, with large neighborhoods, it may also be hard to find functions in $\mathcal{G}_{\text{local}}$ that fit so many points in $S$. Overall, one practical takeaway from this bound is that it is important to not excessively shrink the neighborhood widths if one wants explanations that generalize well for predicting how $f$ behaves at unseen points (see Section 6).

**Caveats.** We remark that this particular bound applies only when the dataset $S$ is solely used to learn $g$; i.e., both $f$ and the neighborhoods must be learned beforehand with separate data[2]. This sort of a framework is typical when deriving theoretical results for models like random forests, where it greatly aids analysis to assume that the trees' splits and their decisions are learned from independent datasets (e.g. two halves of an original dataset) (Arlot & Genuer, 2014). Now, if one is interested in a bound where $S$ is also used to simultaneously learn $f$, one would have to introduce a term corresponding to the complexity of $\mathcal{F}$. Another caveat is that our bound only tells us how well the explanations $g_{x'}$ generalize *on average* over different values of $x'$. This does not tell us anything about the generalization of the quality of $g_{x'}$ for an arbitrary value of $x'$. That being said, just as average accuracy remains a central metric for performance (despite its lack of sensitivity to discrepancies across inputs), average MNF can still be a useful quantity for evaluating an explainer's overall performance.

## 6   EMPIRICAL RESULTS

We present two sets of empirical results to illustrate the the usefulness of our bounds. First, we demonstrate that $\rho_S$ grows much slower than $\mathcal{O}(\sqrt{m})$ which, as stated before, establishes that our bounds yield meaningful convergence rates. Second, we show that Theorem 2 accurately reflects the relationship between explanation generalization (Theorem 2) and the width of $N_x^{\text{mir}}$ used to both generate and evaluate explanations.

**Setup.** For both experiments, we use several regression datasets from the UCI collection (Dua & Graff, 2017) and standardize each feature to have mean 0 and variance 1. We train neural networks as our "black-box" models with the same setup as in (Plumb et al., 2020), using both their non-regularized and ExpO training procedures. The latter explicitly regularizes for NF during training, which we find also decreases MNF on all datasets. For generating explanations, we define $\mathcal{G}_{\text{local}}$ to be linear models and optimize each $g_x$ using the empirical MNF minimizer (see Appendix A). Finally, we approximate $\rho_S$ using a provably accurate sampling-based approach (see Appendix E).

**Growth-rate of $\rho_S$.** In Figure 1 (top), we track the sample dependence of $\rho_S$ for various neighborhoods of width $\sigma$ , setting $N_x^{\text{mir}} = \mathcal{N}(x, \sigma I)$. We specifically approximate the growth rate as polynomial, estimating the exponent by taking the overall slope of a log-log plot of $\rho_S$ over $m$. To cover a natural range for each dataset, $\sigma$ is varied to be between the smallest and half the largest inter-example $l_2$ distances. In these plots, while small $\sigma$ result in a large exponent for $\rho_S$ and large

---

[2]It is due to this reason that we can't plug Theorem 2 into the right hand side of Theorem 1 (in which $f$ depends on $S$) to replace its test MNF term with a train MNF term.

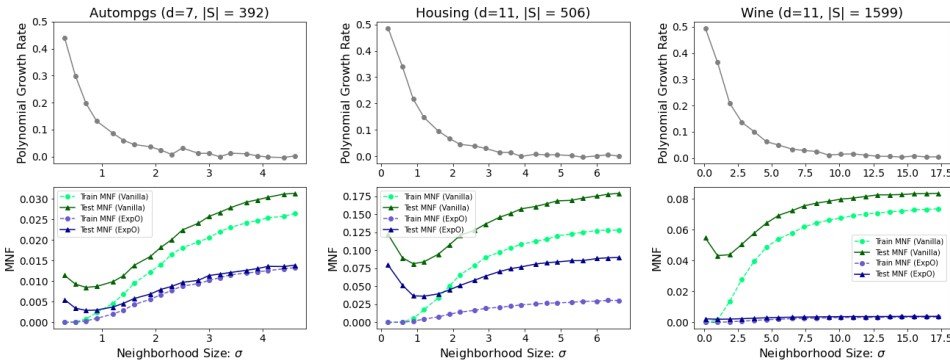

Figure 1: Approximate exponent of $\rho_S$'s polynomial growth rate (top) and train/test MNF (below) plotted for various neighborhood widths across several UCI datasets (see Appendix E for more).

$\sigma$ cause $g$ to intuitively saturate towards a global linear model, we observe that there do exist values of $\sigma$, where both these terms are in control i.e., we observe that we can achieve a growth rate of approximately $\mathcal{O}(m^{0.2})$ without causing $g$ to saturate and MNF metrics to rise sharply.

**Generalization and neighborhood size.** As per the setting of Theorem 2, we generate all explanations using data not used to learn the black-box model. Specifically, we split the original test data into two halves, using only the first half for *explanation training* and the second for *explanation testing*. We plot MNF as measured over these two subsets of examples in Figure 1 (bottom). From the results, it is evident that a generalization gap between train and test MNF exists. Further, recall that Theorem 2 predicts that this gap decreases as wider neighborhoods are used, a phenomena reflected in most of these plots. As a result, while training MNF monotonically increases with larger neighborhoods, test MNF always decreases at certain ranges of $\sigma$.

## 7 CONCLUSION AND FUTURE WORK

In this work, we have studied two novel connections between learning theory and local explanations. We believe these results may be of use in guiding the following directions of future work: (1) developing new local explanation algorithms inspired by our theory and the metric of MNF; (2) resolving caveats or otherwise strengthening the theory presented in this paper; and (3) exploring applications of our techniques beyond interpretability, such as the general problem of deep learning generalization or others that require reasoning about the complexity of randomized functions.

## ACKNOWLEDGMENTS

This work was supported in part by DARPA FA875017C0141, the National Science Foundation grants IIS1705121 and IIS1838017, an Okawa Grant, a Google Faculty Award, an Amazon Web Services Award, a JP Morgan A.I. Research Faculty Award, and a Carnegie Bosch Institute Research Award. Vaishnavh Nagarajan was supported by a grant from the Bosch Center for AI. Any opinions, findings and conclusions or recommendations expressed in this material are those of the author(s) and do not necessarily reflect the views of DARPA, the National Science Foundation, or any other funding agency.

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

# A   MORE ON MIRRORED NEIGHBORHOOD FIDELITY

## A.1   AN ALTERNATIVE INTERPRETATION

Here we elaborate on how the expression for MNF can be parsed in the same way as NF after juggling some terms around. Recall that MNF is defined as:

$$\mathsf{MNF}(f, g) := \mathbb{E}_{x \sim D} \left[ \mathbb{E}_{x' \sim N_x^{\mathrm{mir}}} \left[ (f(x) - g_{x'}(x))^2 \right] \right].$$

and with an abuse of notation, we let $\mathsf{MNF}(f, g, x) = \mathbb{E}_{x' \sim N_x^{\mathrm{mir}}} \left[ (f(x) - g_{x'}(x))^2 \right]$.

Here the outer expectation is over the *target* points $x$ that the explanations try to fit, and the inner expectation is over the *source* points $x'$ which give rise to the explanations $g_{x'}$.

If we can swap these expectations around, we can afford a similar parsing as NF. To get there, first consider the joint distribution over $x$ and $x'$ that is induced by generating $x \sim D$ and then picking $x' \sim N_x^{\mathrm{mir}}$. Under this joint distribution, we need an expression for the marginal distribution of $x'$. This distribution, which we denote by $D^{\dagger}$, is given by:

$$p_{D^{\dagger}}(x') = \int_{\mathcal{X}} p_D(x) p_{N_x^{\mathrm{mir}}}(x') dx.$$

To get a sense of what $D^{\dagger}$ looks like, imagine that $N_x^{\mathrm{mir}}$ is a Gaussian centered at $x$. Then $D^{\dagger}$ corresponds to convolving $D$ with a Gaussian i.e., a smoother version of $D$.

Next, under the same joint distribution, we also need an expression for the distribution of $x$ conditioned on $x'$. This distribution, denoted as $N_{x'}^{\dagger}$, is given by:

$$p_{N_{x'}^{\dagger}}(x) = \frac{p_D(x) p_{N_x^{\mathrm{mir}}}(x')}{\int_{\mathcal{X}} p_D(x) p_{N_x^{\mathrm{mir}}}(x') dx}.$$

Intuitively, $N_{x'}^{\dagger}$ is distribution that is centered around $x'$ and is also weighted by the distribution $D$ i.e., points that are both close to $x'$ and realistic under $D$ have greater weight under $N_{x'}^{\dagger}$. This is because the term $p_{N_x^{\mathrm{mir}}}(x')$ in the numerator prioritizes points that are near $x'$ (imagine $N_x^{\mathrm{mir}}$ being a Gaussian centered at $x$), and the term $p_D(x)$ prioritizes realistic points.

With these definitions in hand, we can swap the expectations around and get:

$$\mathsf{MNF}(f, g) = \mathbb{E}_{x' \sim D^{\dagger}} \left[ \mathbb{E}_{x \sim N_{x'}^{\dagger}} \left[ (f(x) - g_{x'}(x))^2 \right] \right],$$

This then has the same structure as NF in that the outer expectation is over the source points and the inner distribution over target points, and hence can be interpreted similarly. The key insight here is that from the perspective of each individual explanation $g_{x'}$, the relevant distribution it needs to fit (to achieve low MNF) is $N_{x'}^{\dagger}$, which takes into account *both* the local neighborhoods and the real data distribution $D$. In comparison, only the former matters for NF.

## A.2   ALGORITHM FOR MINIMIZING EMPIRICAL MIRRORED NEIGHBORHOOD FIDELITY

We now consider how one might actually fit explanations to minimize MNF. Recall from the above discussion that from the point of view of each source point $x'$, MNF measures how well $g_{x'}$ fits $f$ on the distribution with density $p_{N_{x'}^{\dagger}}(x) = \frac{p_D(x) p_{N_x^{\mathrm{mir}}}(x')}{\int_{\mathcal{X}} p_D(x) p_{N_x^{\mathrm{mir}}}(x') dx}$. Note that one generally does not have access to samples from this distribution due to the dependence on $D$. However as we argue, one can minimize the *empirical* version of MNF given access to a finite sample $S$ drawn i.i.d. from $D$ by solving the following *weighted* regression problem:

$$g_{x'} = \underset{g_{x'} \in \mathcal{G}_{\mathrm{local}}}{\arg\min} \frac{1}{|S|} \sum_{i=1}^{|S|} (f(x_i) - g_{x'}(x_i))^2 p_{N_{x_i}^{\mathrm{mir}}}(x')$$

To see what the empirical version of MNF is, we can replace the outer expectation (over $x \sim D$) in the original definition of MNF with an empirical expectation over $S = \{x_i\}_{i=1}^{|S|}$, giving us

$$\text{Empirical MNF} = \frac{1}{|S|} \sum_{i=1}^{|S|} \mathbb{E}_{x' \sim N_{x_i}^{\text{mir}}} \left[ (f(x_i) - g_{x'}(x_i))^2 \right]$$

$$= \frac{1}{|S|} \sum_{i=1}^{|S|} \int_{\mathcal{X}} (f(x_i) - g_{x'}(x_i))^2 p_{N_{x_i}^{\text{mir}}}(x') dx'$$

$$= \frac{1}{|S|} \int_{\mathcal{X}} \sum_{i=1}^{|S|} (f(x_i) - g_{x'}(x_i))^2 p_{N_{x_i}^{\text{mir}}}(x') dx'$$

To minimize the overall empirical MNF, one can thus choose a $g_{x'}$ for each $x'$ such that it minimizes the above integrand, which is akin to performing one weighted least squares regression. Thus, one notable difference between minimizing empirical MNF and NF is that we need to use real examples to fit $g_{x'}$ for MNF but not for NF since the target distribution of interest there can be user-defined (i.e. it may be chosen such that it can be easily sampled from).

## A.3 TRADE-OFFS BETWEEN MNF AND NF

We now discuss in further detail the comparison between MNF and NF, listing both the relative advantages and disadvantages of each. It should be noted that this discussion is of a somewhat more exploratory nature; we do not aim to make definitive value judgments (i.e. one metric is always more useful than the other), but rather to provide a better qualitative understanding of how these two metrics might be expected to behave. We hope that this discussion prompts a more careful consideration of fidelity metrics in future works involving local explanations.

### A.3.1 ADVANTAGES OF MNF

In many practical situations (esp. for i.i.d. cases), it is reasonable to assume that practitioners will care significantly about generating explanations for predictions at *realistic* on-distribution points and hoping that those (local) models correctly approximate what the model will do at nearby points which are also realistic. Our core argument for the usefulness of MNF compared to NF is that it can be used to come closer to characterizing performance relative to the second part of this goal (i.e. predicting what the model will do at realistic points).

To reiterate Section 3, this is an especially important concern for modern ML settings, which often involve significant feature dependencies (i.e. lower dimensional data manifolds) and models that behave unstably when extrapolating beyond the given task and training data. As we illustrate below in a toy example, when one uses NF with standard neighborhood choices that *don't respect data dependencies* (e.g. $N_x = \mathcal{N}(0, \sigma I)$), one may overemphasize the ability of explanations to fit this noisy behavior on regions that are off-manifold.

**Toy example.** We compare the abilities of MNF and NF to serve as the basis for generating local explanations. In what follows, we refer to $g^{\text{NF}}$ and $g^{\text{MNF}}$ as the *linear* local explainers that minimize NF and MNF respectively. We specifically consider a simple setup where the full input space has dimension $d = 2$ but the data exists on a manifold of dimension $k = 1$. Under task-distribution $D$, let $x_1 \sim \mathcal{N}(0, 1)$ while $x_2 = 0$. Further consider the learned model $f(x) = x_1 - \beta x_1 x_2^2$, where one may assume $\beta \gg 0$. As an important note, on the task distribution $D$, $f$ is functionally equivalent to a fairly simple function, i.e. $f(x) \equiv x_1$.

Minimizing NF: To learn $g_x^{\text{NF}}$, we may simply sample many $x' \sim N_x$ and find a linear $g_x^{\text{NF}}(\cdot)$ that fits these points well. Now, we can expect this process to generalize in a way that $\mathbb{E}_{x' \sim N_x}[(f(x_i) - g_x(x'))^2]$ is minimized. In fact, one could consider the ideal scenario where we sample infinitely many unlabeled examples, and thus find the best possible linear approximation given this neighborhood distribution. However, observe that minimizing the above quantity provides absolutely *no guarantee whatsoever* regarding the error committed on $D$ i.e., $\mathbb{E}_{x' \sim D}[(f(x_i) - g_x(x'))^2]$. This is because $D$ has zero measure when considering $\mathcal{X} = \mathbb{R}^2$. This means that if $f$ is arbitrarily volatile along the irrelevant $x_2$ direction, $g_x$ can also be arbitrarily incorrect on $D$. Indeed, this is the case in the setting above. Letting $g_x(x') = w_1 x_1' + w_2 x_2'$ and $N_x = N(0, I)$, it can be shown that

$\mathsf{NF}(f, g, x)$ is minimized by $w_1 = 1 - \beta$. Since $\beta$ can be arbitrarily large, this explanation can be unboundedly poor at recovering a function equivalent to $f(x) \equiv x_1$ on $D$. Indeed, it can even get the sign of the coefficient for the active feature $x_1$ wrong when $\beta > 1$.

Minimizing $\mathsf{MNF}$: Note that none of the above is a problem when we learn $g^{\mathsf{MNF}}$, because we fit each $g_{x'}^{\mathsf{MNF}}$ only on target points $x$ that are from the real data manifold. [3] This will ensure that $g^{\mathsf{MNF}}$ is in line with a potentially important desiderata for local explanations i.e., that they can faithfully capture a function that is accurate along the task-relevant data directions (of course, only up to a linear approximation). To illustrate more completely, recall that $g^{\mathsf{MNF}}$ is learned as follows: assuming access to $S = \{x_1, \ldots, x_m\} \sim D^m$, we have

$$g_{x'}^{\mathsf{MNF}} = \arg\min_{g_{x'} \in \mathcal{G}_{\text{local}}} \frac{1}{m} \sum_{i=1}^{m} (f(x_i) - g_{x'}(x_i))^2 p_{N_{x_i}^{\text{mir}}}(x')$$

Now since $S$ lies on the manifold of $D$, we have that $x_2 = 0$ on all those points. Therefore, for each $x$, we find the solution which minimizes

$$g_{x'}^{\mathsf{MNF}} = \arg\min_{w_1 \in \mathbb{R}} \frac{1}{m} \sum_{i=1}^{m} (x_{i,1} - w_1 x_{i,1})^2 p_{N_{x_i}^{\text{mir}}}(x')$$

It is easy to see that with just two distinct datapoints from $S$, we would recover $w_1 = 1$, which leads to perfect predictions for how the function behaves on $D$.

As a remark, an even more natural version of the above setting would be one where $f$ is non-linear even on the data manifold. But even here we can still argue that $g^{\mathsf{MNF}}$ would be close to the best possible linear function within the manifold up to a $1/\sqrt{m}$ error (e.g., a generlization bound like our Theorem 2 guarantees this on average over $x$). On the other hand, regardless of how many unlabeled datapoints we fit $g^{\mathsf{NF}}$ with, we would learn $g^{\mathsf{NF}}$ that can behave arbitrarily poorly on the manifold.

### A.3.2 LIMITATIONS OF MNF

Below, we discuss some limitations of $\mathsf{MNF}$ as well as potential future directions for possibly addressing them. At a high-level, we believe while each represents a legitimate concern, they may arguably be (depending on context) "reasonable prices to pay" for the advantages of $\mathsf{MNF}$ compared to $\mathsf{NF}$ described previously.

**MNF explanations may lose local meaning:** Using $\mathsf{MNF}$ to evaluate/generate explanations at low-probability source points $x'$ may have little to do with how $f$ actually behaves around $x'$. Because the target point distribution is $x|x' \propto p_D(x) p_{\mathcal{N}_x}(x')$, very little probability mass might be placed in the vicinity around $x'$ when $p_D(x)$ is small. This would be the case when $x'$ is off-manifold or in low-density regions on the support of the real data distribution. The former might be dismissable if one cares about i.i.d. settings (which is not always the case) but the latter could be very important in applications where rare cases correspond to high-stakes decisions (e.g. disease diagnostics). In these scenarios, the explanation might still be too biased towards how the model is behaving at higher density regions. However, some potential future directions to remedy this are:

- It might help to allow $N_x^{\text{mir}}$ to have smaller width around lower probability points from $D$ (allowing one to concentrate $N_x^{\text{mir}}$ around $x$ despite the form of $D$). It remains a challenge to see how one would actually set these widths but it could be of help if a limit can be assumed on how quickly the value $p_D(x)$ are allowed to change around a given $x$.

- There also could be some utility in considering a more general definition of $\mathsf{MNF}$ that lets one choose an arbitrary outer distribution $x \sim \mathcal{Q}$ other than simply the task distribution $D$. That is, if one really cares about mutually consistent explanations in some arbitrary region (which could be on or off-manifold), then this would potentially allow one to able to customize a metric for that purpose.

---

[3] Recall notation-wise that $\mathsf{NF}$ and $\mathsf{MNF}$ swap the usage of $x$ and $x'$ for target points. Thus, when comparing how individual explanations are generated and evaluated, we refer to them as $g_{x'}^{\mathsf{MNF}}$ and $g_x^{\mathsf{NF}}$ respectively.

**Less intuitive target point neighborhoods:** Very closely related to the previous limitation, in interpreting MNF-based explanations, an end-user would have to understand that $g_{x'}$ are not exactly approximations for the locality around $x'$ but rather for the true target distribution that captures in some sense "on-manifold points near $x'$ (modulated by the concentration of $N_x^{\text{mir}}$)." This makes it harder for a user to know the exact window in which their explanation is directly valid for (compared to a user-specified target neighborhood for NF). In practice, this shortcoming could be at least partially mitigated as long is it is carefully communicated to users that this limitation exists, i.e. they should focus on using MNF explanations only at and for predicting what happens at realistic points.

**Unnaturalness of source points**: While MNF does emphasize realistic target points, it also focuses on explanations generated at potentially off-manifold source points. Further, one could argue that the advantages of MNF are partly because $N_x$ is chosen naively for NF. For instance if one defined $N_x = N_x^\dagger$ in the definition for MNF, then $g^{\text{NF}}$ and $g^{\text{MNF}}$ would produce the same explanations because the relevant (inner) expectations would be the same (comparing NF and the reversed expectation form of MNF from Appendix A.1). However, in this proposal, now the average metric for NF seems more natural in an additional sense since it also only reflects caring about ensuring realistic *source* points (as the outer expectation is over $x \sim D$).

$$\text{NF} = \mathbb{E}_{x \sim D} \mathbb{E}_{x' \sim N_x^\dagger} \left[ [(g_x(x') - f(x')]^2 \right]$$
$$\text{MNF} = \mathbb{E}_{x' \sim D^\dagger} \mathbb{E}_{x \sim N_{x'}^\dagger} \left[ [(g_{x'}(x) - f(x)]^2 \right]$$

Given these two points, might MNF be less interesting on its own? For the first point, we argue that using standard "naive" settings of $N_x$ with NF can also be considered "unnatural" in that it takes into account how explanations at on-manifold source points perform at off-manifold target points. Second, though the specification NF proposed above may be more ideal as a metric, it also becomes less clear how to evaluate it computationally, as the inner distribution cannot be sampled from easily. On the other hand, we can use the original form of writing out MNF (without the expecations flipped) to directly approximate MNF with relevant samples from $D$.

**Inability to reflect what the model "causally" depends on:** In the second toy-example, it was shown that if $f(\mathbf{x}) = x_1 - \beta x_1 x_2^2$ but the data manifold is $(x_1, x_2) = (x_1, 0)$, one could get arbitrarily poor fidelity and feature relevancy (for $x_1$) on this manifold using standard neighborhoods. But MNF runs into a new problem when the feature set actually includes a highly correlated third feature: for example, consider $(x_1, x_2, x_3)$ where the manifold is defined by points $(x_1, x_2, x_3) = (x_1, 0, x_1)$. Thus according to MNF, $g(x) = x_1$, $g(x) = x_3$, and indeed $g(x) = -x_1 + 2x_3$ are all equally good explanations (because MNF only cares about whether $g(x) = f(x)$ on manifold). However, the output of $f$ clearly only "depends" on the input value of $x_1$ for its decisions (in a causal sense). On the other hand, because NF samples target points both on and off manifold, it would correctly see that $x_3$ has no effect. More broadly, the argument here is that in any conversation involving manifolds, one inherently is speaking about some sort of feature dependencies. Thus, one may similarly suffer from the issues of explanations not being causal w.r.t. the output of $f$ and also not being fully identifiable. On the other hand, we note that in the new toy-example, NF is not an ideal fix either because the cost is potentially an arbitrary coefficient for $x_1$ and extremely poor fidelity on $D$. Overall, the intended usage of an explanation plays a large role in whether MNF or NF is indeed appropriate. An argument could be made that finding "what the model uses for its decision", (while perhaps an important goal) is simply not what MNF explanations are trying capture. What one could describe MNF as actually looking at is "can I build a simpler local model relevant to the actual task at hand?"

# B   MORE ON THE DISJOINTEDNESS FACTOR

## B.1   BOUNDS

Recall that the disjointedness factor is defined as $\rho_S := \int_{x' \in \mathcal{X}} \sqrt{\frac{\sum_{j=1}^{m} (p_{N_{x_i}^{\mathrm{mir}}}(x'))^2}{m}} dx'$. Here, we show that it is bounded between 1 and $\sqrt{m}$.

**Fact B.1.** *The disjointedness factor $\rho_S$ satisfies* $1 \leq \rho_S \leq m$.

*Proof.* For the lower bound, we note that since the arithmetic mean lower bounds the quadratic mean, we have:

$$\int_{x' \in \mathcal{X}} \sqrt{\frac{\sum_{j=1}^{m} (p_{N_{x_i}^{\mathrm{mir}}}(x'))^2}{m}} dx' \geq \int_{x' \in \mathcal{X}} \frac{\sum_{j=1}^{m} p_{N_{x_i}^{\mathrm{mir}}}(x')}{m} dx'$$

$$\geq \sum_{j=1}^{m} \frac{1}{m} \int_{x' \in \mathcal{X}} p_{N_{x_i}^{\mathrm{mir}}}(x') dx'$$

$$\geq \sum_{j=1}^{m} \frac{1}{m} = 1$$

For the upper bound, we make use of the fact that the $\ell_2$ norm of a vector is smaller than its $\ell_1$ norm to get:

$$\int_{x' \in \mathcal{X}} \sqrt{\frac{\sum_{j=1}^{m} (p_{N_{x_i}^{\mathrm{mir}}}(x'))^2}{m}} dx' \leq \int_{x' \in \mathcal{X}} \frac{\sum_{j=1}^{m} p_{N_{x_i}^{\mathrm{mir}}}(x')}{\sqrt{m}} dx'$$

$$\leq \sum_{j=1}^{m} \frac{1}{\sqrt{m}} \int_{x' \in \mathcal{X}} p_{N_{x_i}^{\mathrm{mir}}}(x') dx'$$

$$\leq \sum_{j=1}^{m} \frac{1}{\sqrt{m}} = \sqrt{m}$$

$\square$

## B.2   VALUES OF $\rho_S$ IN-BETWEEN 1 AND $\sqrt{m}$

We know that the disjointedness factor $\rho_S$ takes the values 1 and $\sqrt{m}$ in the two extreme cases where the neighborhoods are completely overlapping or disjoint respectively. We also know from Fact B.1 that the only other values it takes lie in between 1 and $\sqrt{m}$. But when does it take these values?

To get a sense of how these in-between values can be realized, we present a toy example. Specifically, we can show that under some simplistic assumptions, $\rho_S = \sqrt{m^{1-k}}$ (where $0 \leq k \leq 1$) if every neighborhood is just large enough to encompass a $\frac{1}{m^{1-k}}$ fraction of mass of the distribution $D$.

Our main assumption is that $N_{x_i}^{\mathrm{mir}}$ is a uniform distribution over whatever support it covers. Further, to simplify the discussion, we assume that $\mathcal{X}$ is a discrete set containing $M$ datapoints in total (think of $M$ as very, very large). Our analysis should extend easily to settings where these assumptions are violated, but we choose to keep these assumptions for readability.

Then, if every neighborhood contains $\frac{1}{m^{1-k}}$ fraction of mass of the distribution $D$, this implies it contains $\frac{M}{m^{1-k}}$ points. Therefore, since $N_{x_i}^{\mathrm{mir}}$ is a uniform distribution, we have that the probability mass of $N_{x_i}^{\mathrm{mir}}$ on any point $x'$ in its support is $\frac{1}{Mm^{k-1}}$. Plugging this in the definition of $\rho_S$, we get:

$$
\rho_S = \int_{x' \in \mathcal{X}} \sqrt{\frac{1}{m} \sum_{i=1}^{m} (p_{N_{x_i}^{\mathrm{mir}}}(x'))^2} dx' = \sum_{j=1}^{M} \sqrt{\frac{1}{m} \sum_{i=1}^{m} \left( \Pr_{x' \sim N_{x_i}^{\mathrm{mir}}} [x' = x_j] \right)^2}
$$

$$
= \sum_{j=1}^{M} \sqrt{\frac{1}{m} \sum_{i=1}^{m} \mathbb{I}[x_j \in \mathrm{supp}\left(N_{x_i}^{\mathrm{mir}}\right)] \left( \frac{1}{Mm^{k-1}} \right)^2}
$$

$$
= \sum_{j=1}^{M} \frac{1}{Mm^{k-0.5}} \sqrt{\sum_{i=1}^{m} \mathbb{I}[x_j \in \mathrm{supp}\left(N_{x_i}^{\mathrm{mir}}\right)]}
$$

To further simplify this, we need to compute the innermost summation, which essentially corresponds to the number of mirrored neighborhoods that each point belongs to. For simplicity, let's assume that every point belongs to $n$ neighborhoods. To estimate $n$, observe that for each of the $m$ neighborhoods to contain $\frac{M}{m^{1-k}}$ points, and for each of the $M$ points to be in $n$ neighborhoods, we must have:

$$
Mn = m\frac{M}{m^{1-k}}.
$$

Thus, $n = m^k$. Plugging this back in, we get $\rho_S = m^{\frac{1-k}{2}}$.

## C  PIECE-WISE GLOBAL APPROXIMATION

### C.0.1  GENERALIZATION BOUND ASSUMING PIECEWISENESS

We now discuss the Rademacher complexity of a simpler class of local-approximation functions: a class of piecewise-simple functions $g \in \mathcal{G}$ with $K$ pieces. In particular, one can show that the complexity of these functions grows with $K$ as $\sqrt{K}$.

To see why, first let us call the $K$ disjoint regions that $g$ is defined over as $R_1, \ldots, R_K$. Correspondingly, the original training set $S = \{x_i\}_i^m$ can be divided into the subsets $S_1 = \{x_{1,i}\}_{i=1}^{m_1}, \ldots, S_k = \{x_{K,i}\}_{i=1}^{m_K}$ that happen to fall into the respective regions and the pieces of $g$ are $g_1, \ldots, g_K \in \mathcal{G}_{\text{local}}$ are simple functions. Then, one can split the Rademacher complexity over the whole dataset in terms of these subsets, to get:

$$
\begin{aligned}
\hat{\mathcal{R}}_S(\mathcal{G}) &= \mathbb{E}_\sigma \left[ \sup_{g \in \mathcal{G}} \frac{1}{m} \sum_{i=1}^m \sigma_i g(x_i) \right] \\
&= \mathbb{E}_\sigma \left[ \sup_{g \in \mathcal{G}} \sum_{k=1}^K \frac{m_k}{m} \sum_{i=1}^m \frac{1}{m_k} \sigma_i g_j(x_i) \mathbb{I}\{x_i \in S_j\} \right] \\
&= \mathbb{E}_\sigma \left[ \sup_{g \in \mathcal{G}} \sum_{k=1}^K \frac{m_k}{m} \sum_{i=1}^{m_k} \frac{1}{m_k} \sigma_{k,i} g_j(x_{k,i}) \right] \\
&\leq \sum_{k=1}^K \frac{m_k}{m} \mathbb{E}_\sigma \left[ \sup_{g_j \in \tilde{\mathcal{G}}} \frac{1}{m_k} \sigma_{k,i} g_j(x_k) \right] \\
&\leq \sum_{k=1}^K \frac{m_k}{m} \hat{\mathcal{R}}_{S_k}(\mathcal{G}_{\text{local}})
\end{aligned}
$$

Now, assuming each $\hat{\mathcal{R}}_{S_k}(\mathcal{G}_{\text{local}})$ is $\mathcal{O}\left(\frac{1}{\sqrt{m_k}}\right)$, in the worst-case, each subset has the same number of points $m_k = m/K$, the sum in the last expression can be bounded as $\mathcal{O}\left(\sqrt{\frac{K}{m}}\right)$.

## D  PROOFS

Below, we present the full statement and proof of Lemma 4.1 which bounds the Rademacher complexity of $\mathcal{G}$. The main difference between this statement and the version in the main paper is that we replace the Rademacher complexity of $\mathcal{G}_{\text{local}}$ with a slightly more carefully defined version of it defined below:

$$\hat{\mathcal{R}}_S^*(\mathcal{G}_{\text{local}}) := \max_{i \le m} \max_{T \subseteq S, |T|=i} \hat{\mathcal{R}}_T(\mathcal{G}_{\text{local}}) \sqrt{\frac{i}{m}} \tag{1}$$

This quantity is essentially a bound on the empirical Rademacher complexity of $\mathcal{G}_{\text{local}}$ on all possible subsets of $S$, with an appropriate scaling factor.

We note that although this quantity is technically larger than the original quantity namely $\hat{\mathcal{R}}_S(\mathcal{G}_{\text{local}})$, for all practical purposes, it is reasonable to think of $\hat{\mathcal{R}}_S^*(\mathcal{G}_{\text{local}})$ as being identical to $\hat{\mathcal{R}}_S(\mathcal{G}_{\text{local}})$ modulo some constant factor. For example, if we have that for all $h \in \mathcal{G}_{\text{local}}$, $h(x) = w \cdot x$ where $\|w\|_2 \le \alpha$, then one would typically bound $\hat{\mathcal{R}}_S(\mathcal{G}_{\text{local}})$ by $\mathcal{O}\left(\frac{\alpha\sqrt{\sum_{i=1}^m \|x_i\|_2^2/m}}{\sqrt{m}}\right)$. The bound on $\hat{\mathcal{R}}_S^*(\mathcal{G}_{\text{local}})$ however would resolve to $\mathcal{O}\left(\frac{\alpha\sqrt{\max_{i \le m} \|x_i\|_2^2}}{\sqrt{m}}\right)$. Now, as long as we assume that $\|x_i\|$ are all bounded by some constant, both these bounds are asymptotically the same, and have the same $1/\sqrt{m}$ dependence on $m$. Additionally, we also remark that that it is possible to write our results in terms of tighter definitions of $\hat{\mathcal{R}}_S^*(\mathcal{G}_{\text{local}})$, however our statements read much cleaner with the above definition.

**Lemma D.1.** *(full, precise statement of Lemma 4.1)* Let $L(\cdot, y')$ be a c-Lipschitz function w.r.t. $y'$ in that for all $y_1, y_2 \in [-B, B]$, $|L(y_1, y') - L(y_2, y')| \le c|y_1 - y_2|$. Let $S = \{(x_1, y_1), \ldots, (x_m, y_m)\} \in (\mathcal{X} \times \mathcal{Y})^m$. Then, the empirical Rademacher complexity of $\mathcal{G}$ under the loss function $L$ is defined and bounded as:

$$\hat{\mathcal{R}}_S(L \circ \mathcal{G}) := \mathbb{E}_{\vec{\sigma}}\left[\sup_{g \in \mathcal{G}} \frac{1}{m} \sum_i^m \sigma_i \mathbb{E}_{x' \sim N_{x_i}^{\text{mir}}}[L(g_{x'}(x_i), y_i)]\right] \le c\rho_S(\ln m + 1) \cdot \hat{\mathcal{R}}_S^*(\mathcal{G}_{\text{local}}).$$

*where recall that* $\rho_S := \int_{x' \in \mathcal{X}} \sqrt{\frac{\sum_{j=1}^m (p_{N_{x_i}^{\text{mir}}}(x'))^2}{m}} dx'$ *is the disjointedness factor.*

Our high level proof idea is to first construct a distribution $\tilde{D}$ over $\mathcal{X}$ such that each of the inner expectations over $N_{x_i}^{\text{mir}}$ (for each $i$) can be rewritten as an expectation over $x' \sim \tilde{D}$. This removes the dependence on $i$ from this expectation, which then allows us to pull this expectation all the way out. This further allows us to take each $x'$ and compute a Rademacher complexity corresponding to the loss of a single local explanation function $g_{x'}$, and then finally average that complexity over $x' \sim \tilde{D}$.

*Proof.* We begin by noting that the inner expectations in the Rademacher complexity are over $m$ unique distributions $N_{x_i}^{\text{mir}}$. our first step is to rewrite these expectations in a way that they all apply on the same distribution. Let us call this distribution $\tilde{D}$ and define what it is later. As long as $\tilde{D}$ has a support that contains the supports of the above $m$ distributions, we can write:

$$\hat{\mathcal{R}}_S(L \circ \mathcal{G}) = \mathbb{E}_{\vec{\sigma}}\left[\sup_{g \in \mathcal{G}} \frac{1}{m} \sum_i^m \sigma_i \mathbb{E}_{x' \sim \tilde{D}}\left[L(g_{x'}(x_i), y_i) \frac{p_{N_{x_i}^{\text{mir}}}(x')}{p_{\tilde{D}}(x')}\right]\right]$$

this allows us to pull the inner expectation out in front of the summation and then the supremum (which now results in an inequality):

$$\le \mathbb{E}_{\vec{\sigma}}\left[\mathbb{E}_{x' \sim \tilde{D}}\left[\sup_{g \in \mathcal{G}} \frac{1}{m} \sum_i^m \sigma_i L(g_{x'}(x_i), y_i) \frac{p_{N_{x_i}^{\text{mir}}}(x')}{p_{\tilde{D}}(x')}\right]\right]$$

which further allows us rewrite the supremum to be over $\mathcal{G}_{\text{local}}$ instead of $\mathcal{G}$:

$$\leq \mathbb{E}_{\vec{\sigma}} \left[ \mathbb{E}_{x' \sim \tilde{D}} \left[ \sup_{h \in \mathcal{G}_{\text{local}}} \frac{1}{m} \sum_i^m \sigma_i L(h(x_i), y_i) \frac{p_{N_{x_i}^{\text{mir}}}(x')}{p_{\tilde{D}}(x')} \right] \right]$$

and finally, let us simply interchange the two outer expectations to rewrite the expression as:

$$\leq \mathbb{E}_{x' \sim \tilde{D}} \left[ \mathbb{E}_{\vec{\sigma}} \left[ \sup_{h \in \mathcal{G}_{\text{local}}} \frac{1}{m} \sum_i^m \sigma_i L(h(x_i), y_i) \frac{p_{N_{x_i}^{\text{mir}}}(x')}{p_{\tilde{D}}(x')} \right] \right].$$

What we now have is an inner expectation which boils down to an empirical Rademacher complexity for a fixed $x'$, and an outer expectation that averages this over $x' \sim \tilde{D}$. For the rest of the discussion, we will fix $x'$ and focus on bounding the inner term. For convenience, let us define $w_i := \frac{p_{N_{x_i}^{\text{mir}}}(x')}{p_{\tilde{D}}(x')}$. Without loss of generality, assume that $w_1 \leq w_2 \leq \ldots \leq w_m$. Also define $w_0 := 0$. We then begin by expanding $w_i$ into a telescopic summation:

$$\mathbb{E}_{\vec{\sigma}} \left[ \sup_{h \in \mathcal{G}_{\text{local}}} \frac{1}{m} \sum_{i=1}^m \sigma_i L(h(x_i), y_i) w_i \right] = \mathbb{E}_{\vec{\sigma}} \left[ \sup_{h \in \mathcal{G}_{\text{local}}} \frac{1}{m} \sum_{i=1}^m \sigma_i L(h(x_i), y_i) \sum_{j=1}^i (w_j - w_{j-1}) \right]$$

then, we interchange the two summations while adjusting their limits appropriately:

$$= \mathbb{E}_{\vec{\sigma}} \left[ \sup_{h \in \mathcal{G}_{\text{local}}} \frac{1}{m} \sum_{j=1}^m \sum_{i=j}^m \sigma_i L(h(x_i), y_i)(w_j - w_{j-1}) \right]$$

and we pull the outer summation out in front of the supremum and expectation, making it an upper bound:

$$\leq \sum_{j=1}^m \mathbb{E}_{\vec{\sigma}} \left[ \sup_{h \in \mathcal{G}_{\text{local}}} \frac{1}{m} \sum_{i=j}^m \sigma_i L(h(x_i), y_i)(w_j - w_{j-1}) \right].$$

Intuitively, the above steps have executed the following idea. The Rademacher complexity on the LHS can be thought of as involving a dataset with weights $w_1, w_2, \ldots, w_m$ given to the losses on each of the $m$ datapoints. We then imagine decomposing this "weighted" dataset into multiple weighted datasets while ensuring that the weights summed across these datasets equal $w_1, w_2, \ldots, w_m$ on the respective datapoints. Then, we could compute the Rademacher complexity for each of these datasets, and then sum them up to get an upper bound on the complexity corresponding to the original dataset.

The way we decomposed the datasets is as follows: first we extract a $w_1$ weight out of all the $m$ data points (which is possible since it's the smallest weight), giving rise to a dataset of $m$ points all with equal weights $w_1$. What remains is a dataset with weights $0, w_2 - w_1, w_3 - w_1, \ldots, w_m - w_1$. From this, we'll extract a $w_2 - w_1$ weight out of all but the first data point to create a dataset of $m - 1$ datapoints all equally weighted as $w_2 - w_1$. By proceeding similarly, we can generate $m$ such datasets of cardinality $m$, $m - 1$, …, 1 respectively, such that all datasets have equally weighted points, and weights that follow the sequence $w_1 - w_0, w_2 - w_1, \ldots$ and so on. As stated before, we will eventually sum up Rademacher complexity terms computed with respect to each of these datasets.

Now, we continue simplifying the above term by pulling out $(w_j - w_{j-1})$ since it is only a constant:

$$\mathbb{E}_{\vec{\sigma}} \left[ \sup_{h \in \mathcal{G}_{\text{local}}} \frac{1}{m} \sum_{i=1}^m \sigma_i L(h(x_i), y_i) w_i \right] \leq \sum_{j=1}^m (w_j - w_{j-1}) \mathbb{E}_{\vec{\sigma}} \left[ \sup_{h \in \mathcal{G}_{\text{local}}} \frac{1}{m} \sum_{i=j}^m \sigma_i L(h(x_i), y_i) \right]$$

next, we apply the standard contraction lemma (Lemma D.2) to make use of the fact $h(x_i)$ is composed with a $c$-Lipschitz function to get:

$$\leq c \sum_{j=1}^{m} (w_j - w_{j-1}) \mathbb{E}_{\vec{\sigma}} \left[ \sup_{h \in \mathcal{G}_{\text{local}}} \frac{1}{m} \sum_{i=j}^{m} \sigma_i h(x_i) \right]$$

using $S_{j:m}$ to denote the datapoints indexed from $j$ to $m$, we can rewrite this in short as:

$$\leq c \sum_{j=1}^{m} (w_j - w_{j-1}) \frac{m+1-j}{m} \hat{\mathcal{R}}_{S_{j:m}}(\mathcal{G}_{\text{local}})$$

and finally, we make use of the definition of $\mathcal{R}_S^*(\mathcal{G}_{\text{local}})$ in Equation 1 to get:

$$\leq c \sum_{j=1}^{m} (w_j - w_{j-1}) \frac{\sqrt{m+1-j}}{\sqrt{m}} \hat{\mathcal{R}}_S^*(\mathcal{G}_{\text{local}}).$$

What remains now is to simplify the summation over $w$'s. To do this, we rearrange the telescopic summation as follows:

$$\sum_{j=1}^{m} (w_j - w_{j-1}) \sqrt{m+1-j} = \sum_{j=1}^{m} w_j (\sqrt{m+1-j} - \sqrt{m-j})$$

$$= \sum_{j=1}^{m} w_j \cdot \frac{1}{\sqrt{m+1-j} + \sqrt{m-j}}$$

$$\leq \sum_{j=1}^{m} w_j \frac{1}{\sqrt{m+1-j}}$$

$$\leq \sqrt{\sum_{j=1}^{m} w_j^2} \cdot \sqrt{\sum_{j=1}^{m} \frac{1}{j}}$$

$$\leq \sqrt{\sum_{j=1}^{m} w_j^2 \cdot (\ln m + 1)}$$

Note that in the penultimate step we've used the Cauchy-Schwartz inequality and in the last step, we have made use of the standard logarithmic upper bound on the $m$-th harmonic number. Plugging this back on the Rademacher complexity bound, we get:

$$\hat{\mathcal{R}}_S(L \circ \mathcal{G}) \leq \mathbb{E}_{x' \sim \tilde{D}} \left[ c \sqrt{\sum_{j=1}^{m} w_j^2 \cdot (\ln m + 1)} \cdot \frac{\hat{\mathcal{R}}_S^*(\mathcal{G}_{\text{local}})}{\sqrt{m}} \right]$$

plugging in the values of $w_j$, we get:

$$\leq \mathbb{E}_{x' \sim \tilde{D}} \left[ c \sqrt{\frac{\sum_{j=1}^{m} (p_{N_{x_i}^{\text{mir}}}(x'))^2}{(p_{\tilde{D}}(x'))^2}} \cdot (\ln m + 1) \cdot \frac{\hat{\mathcal{R}}_S^*(\mathcal{G}_{\text{local}})}{\sqrt{m}} \right].$$

$$\leq c \mathbb{E}_{x' \sim \tilde{D}} \left[ \sqrt{\frac{\sum_{j=1}^{m} \frac{(p_{N_{x_i}^{\text{mir}}}(x'))^2}{m}}{(p_{\tilde{D}}(x'))^2}} \right] (\ln m + 1) \cdot \hat{\mathcal{R}}_S^*(\mathcal{G}_{\text{local}}).$$

Now we finally set $\tilde{D}$ such that $p_{\tilde{D}}(x') = \frac{\sqrt{\sum_{j=1}^{m} \frac{(p_{N_{x_i}^{\mathrm{mir}}}(x'))^2}{m}}}{\rho_S}$ where $\rho_S$ is a normalization constant such that $\rho_S = \int_{x' \in \mathcal{X}} \sqrt{\sum_{j=1}^{m} \frac{(p_{N_{x_i}^{\mathrm{mir}}}(x'))^2}{m}} dx'$. Then, the above term would simplify as:

$$\hat{\mathcal{R}}_S(L \circ \mathcal{G}) \leq c \mathbb{E}_{x' \sim \tilde{D}}\left[\rho_S\right](\ln m + 1) \cdot \hat{\mathcal{R}}_S^*(\mathcal{G}_{\mathrm{local}})$$
$$\leq c\rho_S(\ln m + 1) \cdot \hat{\mathcal{R}}_S^*(\mathcal{G}_{\mathrm{local}}).$$

$\square$

Next, we state and prove the full version of Theorem 1 which provided a generalization guarantee for the test error of $f$ in terms of its local interpretability. This result follows by applying the previous lemma for the squared error loss. For this we need to show that the squared error loss is Lipschitz, which follows from our assumption that the range of the functions in $\mathcal{F}$ and $\mathcal{G}_{\mathrm{local}}$ and also the labels $y$ are bounded in $[-B, B]$.

**Theorem 3.** *(full, precise version of Theorem 1) With probability over $1 - \delta$ over the draws of $S = \{(x_1, y_1), \ldots, (x_m, y_m)\} \sim D^m$, for all $f \in \mathcal{F}$ and for all $g \in \mathcal{G}$, we have (ignoring $\ln 1/\delta$ factors):*

$$\mathbb{E}_{(x,y)\sim D}[(f(x) - y)^2] \leq \frac{4}{m} \sum_{i=1}^{m} (f(x_i) - y_i)^2 + 2 \underbrace{\mathbb{E}_{x\sim D}[\mathbb{E}_{x'\sim N_x^{\mathrm{mir}}}\left[(f(x) - g_{x'}(x))^2\right]]}_{\mathsf{MNF}(f,g)}$$

$$+ \frac{4}{m} \sum_{i=1}^{m} \underbrace{\mathbb{E}_{x'\sim N_x^{\mathrm{mir}}}\left[(f(x_i) - g_{x'}(x_i))^2\right]}_{\mathsf{MNF}(f,g,x_i)} + 16B\rho_S \hat{\mathcal{R}}_S^*(\mathcal{G}_{\mathrm{local}})(\ln m + 1)$$

$$+ 2\sqrt{\frac{\ln 1/\delta}{m}},$$

*where $\rho_S$ denotes the **disjointedness factor** defined as $\rho_S := \int_{x' \in \mathcal{X}} \sqrt{\frac{1}{m} \sum_{i=1}^{m} (p_{N_{x_i}^{\mathrm{mir}}}(x'))^2} dx'$ and $\hat{\mathcal{R}}_S^*(\mathcal{G}_{\mathrm{local}})$ is defined in Equation 1.*

*Proof.* First, we split the test error into two terms by introducing the $g$ function as follows:

$$\mathbb{E}_{(x,y)\sim D}[(f(x) - y)^2] = \mathbb{E}_{(x,y)\sim D}[\mathbb{E}_{x'\sim N_x^{\mathrm{mir}}}[(f(x) - y)^2]]$$
$$\leq 2\left(\mathbb{E}_{x\sim D}[\mathbb{E}_{x'\sim N_x^{\mathrm{mir}}}[(f(x) - g_{x'}(x))^2]] + \mathbb{E}_{x\sim D}[\mathbb{E}_{x'\sim N_x^{\mathrm{mir}}}[(g_{x'}(x) - y)^2]]\right)$$
$$(2)$$

In the first step, we have introduced a dummy expectation over $x'$, and in the next step, we have used the following inequality: for any $a, b, c \in \mathbb{R}$, $(a - b)^2 \leq (|a - c| + |c - b|)^2 \leq 2(|a - c|^2 + |c - b|^2)$ (the first inequality in this line is the triangle inequality and the second inequality is the root mean square inequality).

The first term on the RHS above is $\mathsf{MNF}(f, g)$. To simplify the second term, we first apply a generalization bound based on Rademacher complexity. Specifically, we have that w.h.p $1 - \delta$ over the draws of $S$, for all $g \in \mathcal{G}$,

$$\mathbb{E}_{x\sim D}[\mathbb{E}_{x'\sim N_x^{\mathrm{mir}}}[(g_{x'}(x) - y)^2]] \leq \frac{1}{m} \sum_{i=1}^{m} \mathbb{E}_{x'\sim N_x^{\mathrm{mir}}}[(g_{x'}(x_i) - y_i)^2] + 2\hat{\mathcal{R}}_S(\mathcal{G}) + \sqrt{\frac{\ln 1/\delta}{m}} \quad (3)$$

Now, $\hat{\mathcal{R}}_S(\mathcal{G})$ can be bounded using Lemma 4.1 under Lipschitzness of the squared error loss. Specifically, we have that for $h, h' \in \mathcal{G}_{\mathrm{local}}$, and for all $y \in [-B, B]$, $|(h(x) - y)^2 - (h'(x) - y)^2| \leq$

$4B|h(x) - h'(x)|$, since all of $h(x), h'(x)$ and $y$ lie in $[-B, B]$. Therefore, from Lemma 4.1 we have that:

$$\hat{\mathcal{R}}_S(\mathcal{G}) \leq 4B(\ln m + 1)\rho_S \hat{\mathcal{R}}_S^*(\mathcal{G}_{\text{local}}). \tag{4}$$

The only term that remains to be bounded is the first term on the RHS. This can bounded again using the inequality that for any $a, b, c \in \mathbb{R}$, $(a - b)^2 \leq (|a - c| + |c - b|)^2 \leq 2(|a - c|^2 + |c - b|^2)$:

$$\frac{1}{m}\sum_{i=1}^m \mathbb{E}_{x' \sim N_{x_i}^{\text{mir}}}[(g_{x'}(x_i) - y_i)^2)] \leq \frac{2}{m}\sum_{i=1}^m \mathbb{E}_{x' \sim N_{x_i}^{\text{mir}}}[(g_{x'}(x_i) - f(x_i))^2] + \frac{2}{m}\sum_{i=1}^m (f(x_i) - y_i)^2 \tag{5}$$

By combining the above three chains of inequalities, we get the final bound. $\qquad\square$

Below, we present an alternative version of Theorem 1 where the generalization bound does not involve the test MNF and hence does not require any unlabeled data from $D$; however the bound is not on the test error of $f$ but the test error of $g$.

**Theorem 4.** *(an alternative version of Theorem 1) With probability over $1 - \delta$ over the draws of $S = \{(x_1, y_1), \ldots, (x_m, y_m)\} \sim D^m$, for all $f \in \mathcal{F}$ and for all $g \in \mathcal{G}$, we have:*

$$\mathbb{E}_{(x,y) \sim D}[\mathbb{E}_{x' \sim N_x^{\text{mir}}}[(g_{x'}(x) - y)^2]] \leq \frac{2}{m}\sum_{i=1}^m (f(x_i) - y_i)^2 + \frac{2}{m}\sum_{i=1}^m \underbrace{\mathbb{E}_{x' \sim N_x^{\text{mir}}}\left[(f(x_i) - g_{x'}(x_i))^2\right]}_{\text{MNF}(f, g, x_i)}$$

$$+ 8B\rho_S \hat{\mathcal{R}}_S(\mathcal{G}_{\text{local}})(\ln m + 1) + \sqrt{\frac{\ln 1/\delta}{m}}.$$

*Proof.* The proof follows directly from the proof of Theorem 3 starting from Equation 3. $\qquad\square$

We now state and prove the full version of Theorem 2 which provided a generalization guarantee for the quality of explanations.

**Theorem 5.** *(full, precise statement of Theorem 2) For a fixed function $f$, with high probability $1 - \delta$ over the draws of $S \sim D^m$, for all $g \in \mathcal{G}$, we have:*

$$\underbrace{\mathbb{E}_{x \sim D}\left[\mathbb{E}_{x' \sim N_x^{\text{mir}}}\left[(f(x) - g_{x'}(x))^2\right]\right]}_{\text{Test MNF } i.e., \text{ MNF}(f, g)} \leq \underbrace{\frac{1}{m}\sum_{i=1}^m \mathbb{E}_{x' \sim N_x^{\text{mir}}}\left[(f(x_i) - g_{x'}(x_i))^2\right]}_{\text{Train MNF}}$$

$$+ 8B\rho_S \hat{\mathcal{R}}_S(\mathcal{G}_{\text{local}}) \ln m + \sqrt{\frac{\ln 1/\delta}{m}}.$$

*where $\hat{\mathcal{R}}_S^*(\mathcal{G}_{\text{local}})$ is defined in Equation 1.*

*Proof.* For this result, we need to think of $f$ as a fixed labeling function since it is independent of the dataset $S$ that is used to train $g$. Then, one can apply a standard Rademacher complexity bound and invoke Lemma 4.1 to get the final result (as invoked in Equation 4). $\qquad\square$

Below, we state the standard contraction lemma for Rademacher complexity (e.g., Shalev-Shwartz & Ben-David (2014) Lemma 26.9). The lemma states that composing a function class with a $c$-Lipschitz function can scale up its Rademacher complexity by a multiplicative factor of at most $c$.

**Lemma D.2.** *(Contraction lemma) For each $i = 1, 2, \ldots, m$, let $\phi_i : \mathbb{R} \to \mathbb{R}$ be a $c$-Lipschitz function in that for all $t, t' \in \mathcal{B} \subseteq \mathbb{R}$, $|\phi_i(t) - \phi_i(t')| \leq |t - t'|$. Then, for any class $\mathcal{H}$ of functions $h : \mathbb{R} \to \mathcal{B}$, we have:*

$$\mathbb{E}_{\vec{\sigma}}\left[\sum_{i=1}^m \sigma_i \phi_i(h(x_i))\right] \leq c\mathbb{E}_{\vec{\sigma}}\left[\sum_{i=1}^m \sigma_i(h(x_i))\right].$$

# E EXPERIMENT DETAILS

## E.1 PROCEDURE FOR CALCULATING $\rho_S$

As a reminder, we define $\rho_S$ to be an integral over $\mathcal{X}$, which is not trivial to evaluate in practice, especially when dealing with higher dimensional variables.

$$\rho_S = \int_{x' \in \mathcal{X}} \sqrt{\frac{1}{m} \sum_{i=1}^{m} (p_{N_{x_i}^{\mathrm{mir}}}(x'))^2} dx'$$

Common numerical integration techniques usually incur significant computational costs due to the dimension of $x$. Though a variety of methods exist, one can intuit the inherent difficulty that causes this blow-up by considering the naive approach of simply constructing a Riemann sum across a rectangular meshgrid of points for a $d$-dimensional $\mathcal{X}$. If one wants to create a grid of $c$ points per dimension, then $c^d$ points (and thus evaluations of the integrand) must be processed.

Instead, we can apply Monte-Carlo Integration to evaluate $\rho_S$. As we will see, a key feature of this approach is that error will *not* scale with data dimension and can be bounded probabilistically via a basic Hoeffding bound. Currently, the integral does not take the form of an expectation so we must introduce a dummy distribution $q(x')$ as follows

$$\rho_S = \int_{x' \in \mathcal{X}} \frac{\sqrt{\frac{1}{m} \sum_{i=1}^{m} (p_{N_{x_i}^{\mathrm{mir}}}(x'))^2}}{q(x')} q(x') dx' = \mathbb{E}_{x' \sim q} \left[ \frac{\sqrt{\frac{1}{m} \sum_{i=1}^{m} (p_{N_{x_i}^{\mathrm{mir}}}(x'))^2}}{q(x')} \right]$$

Now, we can estimate $\rho_S$ with $n$ independent samples from $q$.

$$\hat{\rho}_{S,n} = \frac{1}{n} \sum_{j=1}^{n} \frac{\sqrt{\frac{1}{m} \sum_{i=1}^{m} (p_{N_{x_i}^{\mathrm{mir}}}(x_j'))^2}}{q(x_j')}$$

This is obviously an unbiased estimate of $\rho_S$, but that in itself is not sufficient. It is only a feasible approach if we can choose $q$ such that (1) we can actually sample from it, (2) we can calculate $q(x')$ for arbitrary $x'$ and (3) we can control the variance of $\frac{\sqrt{\frac{1}{m} \sum_{i=1}^{m} (p_{N_{x_i}^{\mathrm{mir}}}(x'))^2}}{q(x')}$.

It can be shown by choosing $q$ to be a uniform mixture of the $m$ training set neighborhoods (one for each $x_i \in S$), we can satisfy all 3 properties. (1) and (2) are trivial if those same properties being satisfied by $N_x^{\mathrm{mir}}$ (which is the case for the Gaussian neighborhoods we consider). If $N_x^{\mathrm{mir}}$ can be sampled from, the mixture over $m$ such distributions can obviously be sampled from. The same goes for calculating the density, which in this case is:

$$q(x') = \sum_{i=1}^{m} \frac{1}{m} \cdot p_{N_{x_i}^{\mathrm{mir}}}(x') = \frac{1}{m} \sum_{i=1}^{m} p_{N_{x_i}^{\mathrm{mir}}}(x')$$

We observe that (3) can also be shown because we can upper and lower bound the quantity in question. To show this, we first re-write it as

$$\frac{\sqrt{\frac{1}{m} \sum_{i=1}^{m} (p_{N_{x_i}^{\mathrm{mir}}}(x'))^2}}{q(x')} = \frac{\sqrt{\frac{1}{m} \sum_{i=1}^{m} (p_{N_{x_i}^{\mathrm{mir}}}(x'))^2}}{\frac{1}{m} \sum_{i=1}^{m} \cdot p_{N_{x_i}^{\mathrm{mir}}}(x')}$$

$$= \sqrt{m} \frac{\sqrt{\sum_{i=1}^{m} (p_{N_{x_i}^{\mathrm{mir}}}(x'))^2}}{\sum_{i=1}^{m} \cdot p_{N_{x_i}^{\mathrm{mir}}}(x')}$$

$$= \sqrt{m} \frac{||p_S(x')||_2}{||p_S(x')||_1}$$

where $p_S(x')$ is a $m$-dimensional vector of densities each evaluated at $x'$ (i.e. one for each of the $m$ training points). Since for any vector $v$, $||v||_2 \le ||v||_1 \le \sqrt{m}||v||_2$, the upper and lower bounds for this quantity in question are $\sqrt{m}$ and 1 respectively. Thus we can bound the variance of this quantity by $\frac{1}{4}(\sqrt{m}-1)^2 \le \frac{m}{4}$ and $\text{Var}(\hat{\rho}_{S,n}) \le \frac{m}{4n}$. This does not scale with dimension but only the number of training points!

To be even more concrete, for a given $m$ and $n$, we can now apply a Hoeffding bound to control the error.

$$\mathbb{P}(|\hat{\rho}_{S,n} - \rho_S| > t) \le 2e^{\frac{-2nt^2}{m}}$$

In our experiments we choose $n$ to be $10m$, meaning that the probability that $\rho_S$ is off by more than 0.5 is capped at about 1% (recall that $\rho_S$ scales from $[1, \sqrt{m}]$, which means this is a fairly reasonable degree of accuracy).

## E.2 FULL SET OF RESULTS

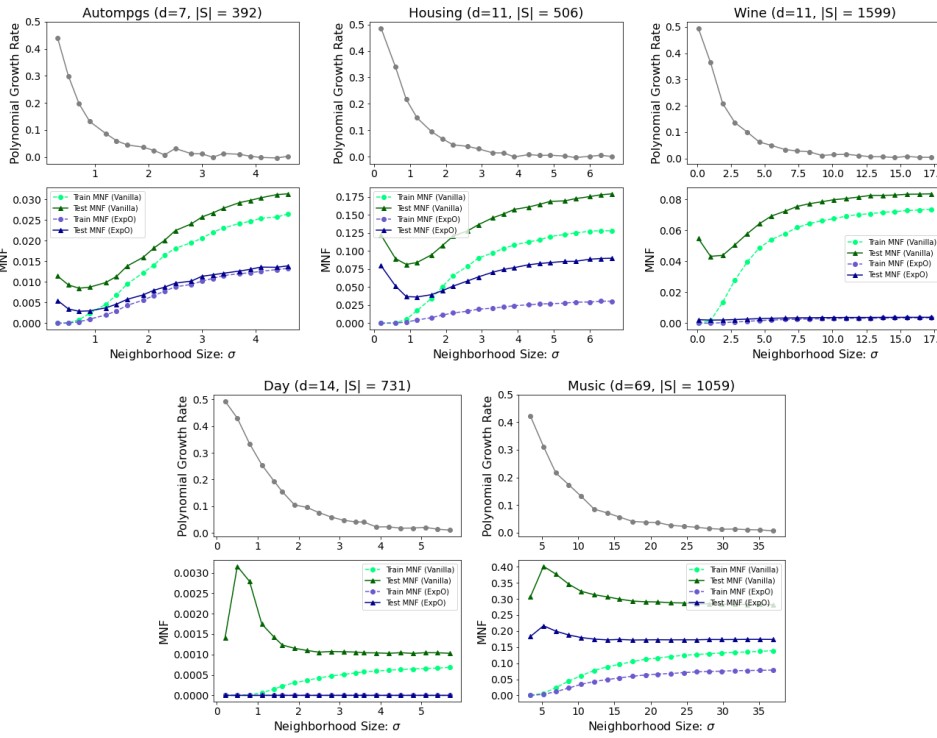

Figure 2: Approximate exponent of $\rho_S$'s polynomial growth rate (top) and train/test MNF (below) plotted for various neighborhood widths across several UCI datasets.

Note that for the *Day* and *Music* datasets, the test MNF curves behave differently than for the other three datasets. Here, test MNF rises initially for small neighborhood widths but then drops and plateaus as neighborhoods get larger. As noted by Plumb et al. (2020), *Day* exhibits (globally) a fairly linear relationship between inputs and outputs. Thus, we hypothesize our results here can be explained on the basis that a global linear model (which is what the explanations saturate towards as $\sigma$ increases) actually can do quite well at approximating $f$. That is, in the terms of the bound, the train MNF does not rise too sharply when saturation occurs but the complexity terms become smaller as neighborhoods get wider. We hypothesize a similar effect is at play for *Music*.

