# OpenReview forum: "A Learning Theoretic Perspective on Local Explainability"
_ICLR.cc/2021/Conference — ICLR 2021 Poster_

### Official Review · AnonReviewer2 · 2020-10-25

**Rating:** 7
**Confidence:** 3

**Review:**

Summary:

This paper presents two main theoretical results as its main contributions.  First, the authors provide a bound on the model generalization in terms its local explainability.  This bound relates the model generalization, the training accuracy, local explainability, and the complexity of the explanations.  Second the authors provide a bound on local explainability generalization.  This bound describes the relation between training local explainability and test time local explainability.  Both these bounds make concrete intuitive notions around local explainability generalization --- for instance, as we increase the size of the local explanation, generalization will decrease.  Last, the authors provide short empirical evaluation to demonstrate the properties they describe in their bounds appear in practice.

Questions + Comments:

Section 3:

Both notions of neighborhood fidelity (NF) and mirrored neighborhood fidelity (MNF) are well described.  As the authors comment, NF is the commonly used notion of explanation fidelity --- the local error on sampled points within the neighborhood.  The authors perform their analysis on the related notion MNF, which relies on evaluating instead on training points. The authors comment that this notion is (1) relevant and well motivated (2) enables the connection between generalization and local interpretability.  While there is likely a good reason, its currently unclear why the same analysis can't be performed on the more commonly used NF notion.  Right now, the transition from NF and MNF lacks motivation and stating why MNF is needed very explicitly would help make the section much stronger.

Section 4:

The significance and meaning of Rademacher complexity could be more clearly described.  Though this might be a common tool for those from a learning theory background, readers with backgrounds in explainability could be less familiar.  For instance, what is $\sigma_i$ in lemma 4.1? What is the significance of this term? Stating this more clearly within the section could strengthen the applicability to explainability focused readers.

Section 5:

In the second paragraph, the authors claim that notions of local explanations which can repeatedly sample nearby points are not interesting from a generalization perspective because we have access to a potentially unlimited amount of data. However, in these situations, querying the model so much as to reach this level of data might not be practical.  Consider something like BERT where query time could be prohibitively slow (and f is not free by any means, both from a time and $ perspective) --- questions related to how well a particular explanation could generalize could be very useful for things like deciding when to stop querying the model.  I'd ask the authors to at least reconsider this strong assertion that investigation into this direction is not useful.  Further, I'd be curious to hear if the author's results could apply in this case?

Overall comments:

This is a well written paper that outlines a number of useful properties of local explanations.  Because the results generally use the MNF notion of fidelity, some users of local explanations might not find them as immediately applicable.  Further, the results might not be so surprising to interpretability focused readers -- the assertion that decreasing the width of the local explanation will likely result in a better approximation is fairly well understood for instance, at least intuitively.  That said, the paper makes clear a number of key properties in local explanations and thus could be a useful contribution to the literature.

---

> ### Author Response · Authors · 2020-11-18
> **Response to Reviewer 2**
>
> Thank you for your time and thoughtful review! We hope to address some of your questions and comments below, and would welcome any additional comments/questions.
>
> 1.) Why can’t the same analysis be applied to NF?
>
> Thanks for bringing up this point! Indeed, we initially focused on NF, but encountered some fundamental issues in the analysis which led us to discover MNF. At a high-level, our results don’t apply to NF because in NF, the overall distribution of the points that are “fit” to calculate NF (i.e., the target points), is not the same as $D$ (instead being $D$ convolved with $N_x$). However, we want this distribution to be the same as $D$ (which is the case in MNF) for us to be able to neatly bound the test error term via the local interpretability term. Otherwise, we would have to end up introducing many cumbersome terms.  We have incorporated this discussion in the revised version of Sec 3!
>
> 2.) Rademacher Complexity is not explained well for new audiences
>
> Again, we appreciate this point for making our work more accessible to an interpretability audience. We have added this discussion in the current revised version of the paper. Rademacher complexity is the ability of a function class to be able to fit noise. Here, $\sigma_i$ are i.i.d  $\pm 1$ variables which can be thought of noisy labels. The sup tries to find the best fit in the function class.
>
> 3.) Infeasibility of sampling indefinitely
>
> This is an interesting point! Since we had been mainly concerned with the theoretical aspects of the various learning problems involved, we had not considered practical costs of querying too many new examples. For scenarios where one is limited to a budget of finite queries, there are perhaps two cases to consider:
>
> Case 1: In each neighborhood where you learn an explanation, you’re allowed to query a fresh but limited set of points drawn from this neighborhood (e.g. LIME). In this case, our theorem would be an overkill. Here, for each neighborhood, one can apply a “separate” uniform convergence bound for its explanation. Concretely, for any $x$, let’s say you draw a fresh set $S_{x}$ to learn $g_{x}$. Then, you could say that w.h.p. over the draws of $S_{x}$, for all $g_{x} \in \mathcal{G}_{local}$, one can bound the error of $g_x$ on its neighborhood by the error of $g_x$ on $S_x$ and $\mathcal{O}(\mathcal{R}_{S_x}(G_{local}))$. Here observe that the complexity term doesn’t have the $\rho$ term from our more involved analysis of Thm 2, where we had to apply uniform convergence simultaneously over all explanations because we used a common dataset to learn all of them.
>
> Case 2: Across all neighborhoods, you are forced to reuse the same set of points by re-weighting. This is the exact kind of setting for which our Theorem 2 would apply and be relevant.
>
> We acknowledge the current manuscript is probably overly dismissive of Case 1, but our main point is that while there is a generalization question in that scenario, it also can be addressed much more trivially. We have edited our discussion of this point in the revised submission in Sec 5!
>
> 4.) Results depend on MNF so they may not be immediately applicable to local explanation users
>
> This may certainly be true and valid as a direction for future improvement. In the context of Theorem 2, we ourselves also noted a similar point in the subparagraph titled “the indirectness of our result.” In our view, if one specifically cared about about NF instead, this indirectness would perhaps mediated by MNF and NF being similar in form and spirit, with both measuring local interpretability via squared error averaged across neighborhood distributions. However, we overall still see our work as providing value by taking a concrete first step towards connecting generalization and local explainability formally.
>
> 5.) Results may not be surprising to interpretability focused readers
>
> We recognize that the implications of our results tend to be observations that might be intuitive in retrospect. However, we believe this concordance of theory with expectation should in fact be viewed as a positive indication of our results’ validity in the context of our main goal, which was to meaningfully capture relationships between generalization and local explanations. Additionally, while these were not the main focuses of our paper, we also think our theory naturally presents some useful algorithmic ideas for future study, namely (1) further/more rigorous investigation into MNF and its proposed benefits for generating/evaluating explanations; (2) improved sample-dependent (i.e. MAPLE-like) local explanations that dynamically optimize each neighborhood to be as large as possible (given a fidelity “budget”).

---

> > ### Comment · AnonReviewer2 · 2020-11-20
> > **Thanks for the response**
> >
> > I appreciate the responses and extensions to the paper. In regards to points (4) and (5), I agree with the authors that this is a useful direction of study and clearly describing this connection is important, even if it might be intuitive in retrospect as you said.  I'm in favor of accepting this paper and have updated my score to reflect this.

---

### Official Review · AnonReviewer1 · 2020-10-29
**Well written, insightful paper**

**Rating:** 7
**Confidence:** 4

**Review:**

The manuscript proposes bounds for the MSE of the estimation of a function from finite samples. The novelty of the proposed bounds in comparison to classical results is that the complexity of the function is stated in terms of its local interpretability, i.e. how well it can be approximated by a family of simple functions.

The paper is very well written and clear. I think that the proposed way of characterizing function complexities is useful for providing new insights in learning theory. However, the authors may find it useful to address the following points:

1. In the main result of the paper (e.g. the bound in the first page) would it be typically easier to find the complexity of the local approximation function class G, rather than finding the complexity of the actual function class F? Some comments would be useful.

2. The neighborhood fidelity (NF) concept was proposed in some previous works, which the authors have modified to propose the mirrored neighborhood fidelity (MNF). The main difference between these two definitions seems to be the fact that the local approximations gx are evaluated at points x' within the neighborhood of a source point x in NF; while in MNF a different local approximation gx' is computed at each neighboring point, and gx' are evaluated at x. The authors have presented some arguments about the advantages of MNF over NF, but the essential difference between these two ideas is still not clear to me.

a. It is said that "Selecting the target point distribution to be D rather than D perturbed by Nx better emphasizes the ability for explanations to accurately convey how well g will predict at realistic points." In my understanding, both x and x' will be samples in the data set in the empirical estimate of NF, as well as that of MNF. So I do not understand why MNF is more realistic than NF.

b. It is also said that "When one measures NF with standard neighborhood choices that
ignore feature dependencies (i.e. most commonly Nx = N(x, sigma I)), the resulting target distribution may concentrate significantly on regions that are non-relevant to the task at hand". Again, in my understanding the choice of the neighborhood is something independent of whether NF or MNF is used. Both could be coupled with any neighborhood definition.

3. Why don't the authors combine theorems 1 and 2 to bound the expected loss in terms of the empirical MNF and the Rademacher complexity of the local approximations? I guess that would be the most meaningful result from this paper. The main result summary in the first page of the paper does not exactly do this.

4. The authors mention some limitations of their bounds referring to the dependence on the dimensionality. I guess to address this limitation, a nice extension of this study would be to remodel the neighborhood size by taking into account the intrinsic dimensionality of the data, using e.g. manifold models, low-dimensional models, etc. A Gaussian neighborhood seems to be considered throughout the paper. This often gives a too high estimation of the dimensionality of the data, whereas actual data sets are often low-dimensional although they reside in a high-dimensional space.

5. In the practical scenario considered in this work, do gx's have to be computed in practice? I understand that the role of gx's is just to theoretically characterize the function complexity of f, so the proposed bounds would be applicable to any learning algorithm (not necessarily using local approximations), is this correct?

---

> ### Author Response · Authors · 2020-11-18
> **Response to Reviewer 1**
>
> Thank you for your time and thoughtful review! We hope to address some of your questions and comments below, and would welcome any additional comments/questions.
>
> 1.) Typically easier to bound the complexity of $\mathcal{G}$ than $\mathcal{F}$?
>
> Yes, some instances where our theorem would likely be useful are cases in which the complexity of $\mathcal{G}$ is easier to theoretically analyze and/or bounding the complexity of $\mathcal{F}$ just leads to non-useful bounds (e.g. often the case when $\mathcal{F}$ is a class of neural networks). Indeed, in standard "interpretability settings", $\mathcal{G}$ will usually be something much simpler and more amenable to standard analysis (e.g, linear models or decision trees) than $\mathcal{F}$. But even in cases where $\mathcal{F}$ is easy to analyze, our theorem still has value as it provides some intuition in terms of how local explainability relates to generalization. We have added a comment about this in Sec 4!
>
> 2.) MNF vs. NF
>
> The main takeaway is that these are closely related quantities since they measure local interpretability via squared error averaged across neighborhoods. As you note, the main difference is in how the source distributions (i.e. points local explanations are centered on) and target distributions (i.e. points that individual local explanations are evaluated on) are chosen respectively. In NF the source points, $x$, are drawn from the real data distribution D, but the target points, $x’$, are drawn from some data distribution which may overlap with both realistic/unrealistic regions of the input space (e.g. a gaussian centered at x, which ignores important feature dependencies). In MNF, this is flipped and $x$ refers to the target points which are drawn from $D$ while $x’$ refers to source points which are drawn to be conditionally near various $x$.
>
> Thus, in response to your point a., the samples from the neighborhoods ($x’ \sim N_x$) in both expressions may not necessarily be examples from the real dataset. However, NF uses these potentially unrealistic points as target points whereas MNF only considers them as source points. Our arguments for the potential benefit of MNF arise from this discrepancy, as MNF in some sense forces explanations to fit $f$ well on $D$ by focusing only on realistic target points.
>
> As for your point b., it indeed may be possible to choose $N_x$ and $N_x^{mir}$ in such a way to make MNF/NF more similar to each other. However, past practice has been to choose $N_x$ as something that captures a sense of locality while also either being (a) independent of $D$ but allowing one to sample from the $N_x$ indefinitely; (b) dependent on $D$ but where one only has access via a finite sample. The former case (e.g. isotropic Gaussian, uniform over ball) is where our arguments directly hold. The latter case is indeed more similar in spirit to MNF but calculating NF here may not be fully tractable in the way MNF is (see Appendix A.2.2 for an example).
>
> 3.) Combining Theorems 1 and 2?
>
> Theorems 1 and 2 assume different “worlds” where the dataset dependencies of $f$ and $g$ conflict. In Theorem 1, we care about $f$’s generalization over the training set it was learned on, $S$ (i.e. the dataset the empirical terms in the bound are calculated over). Here, $f$ (and subsequently $g$) can depend on $S$. On the other hand, Theorem 2 concerns the generalization of $g$ over a separate training set it was learned on, assuming $f$ is fixed (i.e. learned over a separate dataset). Here, one considers a different $S$ which $g$ depends on but $f$ is independent of. We’ve added a clarification about this at the end of Sec 5!
>
> 4.) Exploiting low intrinsic dimensionality
>
> This is a nice point! Despite usage in prior work, isotropic Gaussian neighborhoods are indeed a very limiting choice for defining $N_x$. In the context of our existing results, which applies generally for any $N_x$, considering intrinsic dimensionality could indeed allow one to choose “better” $N_x$ that don’t place mass off-manifold. Tying back to 2b.), this perhaps could make MNF and NF even more similar to each other. It could also likely lead to tighter bounds for certain datasets, as it might result in less disjoint neighborhoods (reducing $\rho_S$). Since the idea of taking into account intrinsic dimension (as well as perhaps other ideas for smartly choosing $N_x$) has not really been explored in existing literature, we definitely think it would be a novel avenue for future work!
>
> 5.) Does $g_x$ have to be computed? Do bounds apply to any learning algorithm?
>
> Yes, you need to have computed $g_x$ (using any learning algorithm) to be able to  compute the MNF terms in the bounds. Indeed, in the context of interpretability settings, $g_x$ is what is computed to generate explanations, so it is reasonable to assume that they are computed explicitly. Also, all of this can be applied regardless of what algorithm is used to learn $f$ (which can just be considered a black-box).

---

### Official Review · AnonReviewer3 · 2020-11-06
**Connection between local explainability and learning theory**

**Rating:** 5
**Confidence:** 3

**Review:**

This paper tries to make a novel connection between local explainability and learning theory, and proposes two theorems regarding bounds related to performance generalization and explanation generalization, respectively. The paper presents two sets of empirical results to illustrate the the usefulness of our bounds.

Overall, the idea of the paper is interesting by exploring local explanations of black-box machine learning models from aspects of learning theory. Mirrored Neighborhood Fidelity (MNF) is proposed as a novel measure of local explainability and the core component of arguments and conclusions in the paper.

The paper claims that MNF naturally complements commonly used Neighborhood Fidelity (NF) and offers a unique advantage over NF when evaluating local explanations on “realistic” high-dimensional on-distribution data which often exhibit significant feature dependencies. However, there is no solid or convincing proof and empirical experiments for supporting the above claims except a toy example in the appendix.

The experimental section in the paper plots the polynomial growth rate and MNF for various neighborhood widths across several UCI datasets, but doesn’t show how the proposed MNF-based method can gain advantage over existing NF-based local explanation models such as LIME and MAPLE in terms of better explanation.

There is some space to improve readability. Some notations are not well-defined and self-contained for their first mentions.

---

> ### Author Response · Authors · 2020-11-18
> **Response to Reviewer 3**
>
> Thank you for your time and thoughtful review! We hope to address some of your comments below, and would welcome any additional comments/questions.
>
> 1.) Advantages of MNF (compared to NF)
>
> We acknowledge that our discussions of the benefits of MNF vs. NF are more exploratory in nature and would require further proofs/experiments to be more conclusive.  That being said, we do not see the favorability of MNF as the main contribution of our work, which is instead to establish a first meaningful foothold for connecting local explainability and generalization. Towards this goal, since our theory depends on introducing MNF, we mainly aimed to argue that  MNF is a reasonable characterization (i.e. arguably as reasonable as NF) of local explainability. Thus, our discussion of the two metrics focuses mostly on the fact that these are largely closely related quantities since they measure local interpretability via squared error averaged across neighborhood distributions. In response to your concerns, we have updated the intro as well as the last few paragraphs in Section 3. We hope this better reflects our above perspectives!
>
>
> 2.) Readability
>
> Thank you for pointing this out! For the introduction of Rademacher Complexity, we’ve made sure to add a more clear description for defining sigma in the revised submission. We will be careful to re-read, clarify, and list further corrections we find in future revisions.

---

### Decision · Program_Chairs · 2021-01-07
**Final Decision**

**Decision:**

Accept (Poster)

**Comment:**

This paper presents an interesting connection between learning theory and local explainability. The reviewers have reacted to each others' thoughts, as well as the authors' comments; they are largely in favor of acceptance. I think the ICLR community will enjoy discussing this paper at the conference.